# Nutrient Digestive Bypass: Determinants and Associations with Stool Quality in Cats and Dogs

**DOI:** 10.3390/ani14192778

**Published:** 2024-09-26

**Authors:** Matthew I. Jackson, Susan M. Wernimont, Kristen Carnagey, Dennis E. Jewell

**Affiliations:** 1Hill’s Pet Nutrition, Topeka, KS 66603, USA; 2Department of Grain Science and Industry, Kansas State University, Manhattan, KS 66506, USA; djewell@ksu.edu

**Keywords:** cat, dog, macronutrients, digestibility, stool, age, fecal quality

## Abstract

**Simple Summary:**

This study shows that nutrients that are not digested have a significant effect on stool moisture and quality. As protein in the lower gut increased, stool moisture increased and quality decreased in both species. Also, increasing fiber in the lower gut resulted in increased stool quality and reduced moisture in both species. Both species also had a response to advancing age: in the dog, age increased firmness for the first 11 years of life, while in the cat, age increased firmness after 4 years of age.

**Abstract:**

The effect of digestive bypass macronutrients and age on stool quality (moisture and firmness) in dogs and cats is not well understood. Data were analyzed from digestibility tests (*n* = 2020, 361 dogs and 536 cats) including dry and wet product types. Both food and feces were measured for moisture and nutrients according to standard protocols; stool firmness was graded. Linear mixed modeling was used to evaluate the associations between nutrient bypass, age and stool quality. Bypass protein increased stool moisture (dog, cat *p* < 0.0001) and decreased firmness (dog *p* = 0.01, cat *p* < 0.0001), while bypass fiber decreased stool moisture and increased firmness (dog, cat *p* < 0.0001 for both). Both species manifested a negative quadratic effect of advanced age on stool firmness (dog *p* < 0.0001 and cat *p* = 0.02). However, the association of advanced age (quadratic effect) with metabolizable energy required to maintain body weight was different between species; dogs had a positive association (*p* = 0.028), while it was negative for cats (*p* < 0.0001). Taken together, these data may aid in the development of food formulations for companion animals, which can better meet changing nutritional needs across life stages.

## 1. Introduction

The digestion of food is never completely efficient; some portion of dietary nutrients bypasses the upper gastrointestinal (GI) tract to arrive in the colon. These bypass nutrients can directly influence colon physiology and motility. They are also extensively catabolized by resident hindgut bacteria to lower molecular weight products, having impact on host colon and systemic health [1]. Protein, fat and non-digestible carbohydrate (fiber) are subject to microbial proteolysis, lipolysis and saccharolysis, respectively. These are further putrified and fermented by gut microbes, with the final catabolic products impacting two determinants of stool firmness: colonic motility [2] and water content [3]. Additionally, microbial metabolic products of bypass nutrients can influence the inflammatory state of the colon, which may obliquely impact stool firmness [4]. Measures of stool quality can include subjective stool firmness and objective levels of stool moisture. Stool firmness scoring is a subjective, albeit useful, measure of general intestinal health in terms of appropriate colon motility and osmotic balance, as well as the absence of overt inflammation and diarrhea-related pathogenic bacteria. While the impact of fiber on stool quality is well accepted, the degree to which dietary protein and fat bypass digestion and interact with other nutrients to affect stool quality is not well elucidated.

Apparent total tract digestibility (ATTD) is defined as the proportion of foodstuff taken from the digestive tract by absorption. The Association of American Feed Control Officials (AAFCO) outlines two methods for digestibility studies, the quantitative collection method and the indicator method, which pet food companies commonly use when testing the digestibility of their foods. The quantitative collection method is more commonly used and is performed by feeding an investigational diet to the pet and then measuring the proportion of nutrients consumed that are absorbed by the body, calculated as ATTD = (grams consumed − grams in feces)/grams consumed [5]. True total tract digestibility (TTTD) is a modification of ATTD, wherein a quantity of protein or fat is subtracted from the amount observed in feces to account for the fact that some amount of protein or fat observed in feces is due to the loss of these components from the sloughing of endogenous or integument tissues. Because a change in nutrient digestibility results in a change in fecal nutrient concentration, there is a direct relationship between digestibility and the proportion of fecal nutrients from endogenous and integumental losses.

Pet age and the level of nutrient intake have been proposed to be determinants of digestibility and thus to influence the degree to which bypass nutrients are available in the colon. While it is well recognized that nutritional requirements for cats and dogs vary throughout their lifespan [6], changes in macronutrient digestibility that may occur as pets age and the impact of digestibility changes on health parameters such as stool quality are less well understood. Aging is associated with changes in nutrient intake as well as gastrointestinal changes, including alterations in microbiome composition, morphology and immunocompetence, all of which may affect digestibility [6,7,8,9,10]. 

This meta-analysis was conducted to elucidate the associations between indigestible bypass nutrients and stool quality in dogs and cats. Additionally, the roles of age and nutrient intake as determinants of digestibility were examined using what is to date the largest data set compiled to investigate these factors. These findings should improve our understanding of dietary factors which may result in increased bypass nutrients, as well as highlight the changing nutritional needs in dogs and cats in response to observed changes in nutrient digestion as pets age. These results support the goal of achieving a more appropriate nutrient content of food and thus optimizing health for pets throughout their lifespan.

## 2. Materials and Methods

### 2.1. Study Population

All of the cats and dogs included in these studies were healthy. None had chronic systemic disease as determined by an annual physical examination, complete blood counts, serum biochemical analyses, and urinalysis. The dogs included in these studies were immunized against canine distemper, adenovirus, parvovirus, Bordetella, and rabies, and the cats were immunized against rabies, viral rhinotracheitis, feline calicivirus, and feline panleukopenia virus. The dogs were housed individually in indoor runs, with structured group exercise. The cats were housed individually and allowed exercise in indoor runs. Both dogs and cats had access to natural light and were provided with regular opportunities for socialization and environmental enrichment. Dogs and cats experienced behavioral enrichment through daily interaction and play time with caretakers, with access to toys, and by daily run and exercise opportunities for the dogs. The study protocols were reviewed and approved by the Institutional Animal Care and Use Committee, Hill’s Pet Nutrition, Inc., Topeka, KS, USA. 

A total of 2065 quantitative digestibility tests that were initiated between December 1997 and April 2017 using 1924 diets consisting of 12,056 individual animal × test data points were assessed for inclusion in the present analysis. Huber robust outlier rejection criteria were applied to the following endpoints: percent change in body weight during digestibility test, apparent carbohydrate digestibility, apparent dry matter digestibility, apparent energy digestibility, apparent fat digestibility, apparent fiber (crude) digestibility, and true protein digestibility [11]. Outliers were determined on a by-group basis, according to the following four groups: dogs fed dry food, dogs fed wet food, cats fed dry food, cats fed wet food (Appendix A). After iterative Huber robust outlier rejection, 785 individual points were eliminated (6.5% of the initial data) to arrive at 11,271 individual animal × test data points (dog *n* = 5789; cat *n* = 5482) from 2020 tests utilizing 1885 foods (Appendix A). For digestibility studies conducted using healthy dogs (1 to 16 years of age; *n* = 361), there were 798 tests using 744 dry foods (animal × test data points = 4525) and 230 tests using 222 wet foods (animal × test data points = 1264). The removal of the outliers had relatively small changes in the central tendency of the data. Numerically, the change in body weight central tendency was the greatest change. However, as this number approaches zero by design, numerically large changes are not associated with biologically significant changes in the outcome. The dogs were primarily beagles (*n* = 318) with five mixed-breed and 38 small breed dogs. For digestibility studies conducted using healthy cats (6 months to 16 years of age; *n* = 536), there were 648 tests using 593 dry foods (animal × test data points = 3728) and 344 tests using 326 wet foods (animal × test data points = 1754). The cats were primarily domestic short hair (*n* = 525). Most animals participated in more than one digestibility test, and mixed model statistical testing was employed to account for the repeated inclusion of individual animals in the data set. 

### 2.2. Foods

A variety of foods were studied, including dry and canned foods, with varying nutrient compositions intended for growth, adult and senior life stages, and a variety of lifestyles and health statuses. Foods were Hill’s^®^ pet foods (Hill’s Pet Nutrition Inc., Topeka, KS, USA) or other commercial pet food brands obtained from the US, Europe, and South America. Both commercial and non-commercial Hill’s^®^ pet foods were tested. All foods complied with canine or feline AAFCO requirements for complete and balanced pet foods [5]. Because some foods complied by passing an AAFCO feeding test, macronutrient compositions which fell outside of the minimum on the relevant AAFCO nutrient profile were able to be included in this assessment.

### 2.3. Study Design and Measurements

All digestibility studies followed the AAFCO protocols established at the time of the study as summarized in the AAFCO Official Publication for that year, using the quantitative collection protocol. All animals were fed to maintain body weight. Each test included 6 adult dogs or cats and consisted of 2 phases [5]. The first phase was a pre-collection period of at least 5 days to allow the acclimation of the animals to the test food and to adjust food intake, as needed, to maintain body weight. The second phase lasted 5 days and involved the total collection of feces. The amount of food offered during the second phase was held constant and based upon the amount of food that maintained body weight in the first phase. Water was freely available to animals at all times. 

Analytical analyses for protein, fat, moisture, fiber, ash (a measure of mineral content) and energy in food and feces were performed as outlined by the AAFCO [5]. Total fiber intake was measured as crude fiber according to the AOAC method 962.09. Digestibility coefficients for dry matter, nitrogen-free extract (NFE; digestible carbohydrate) and fiber were calculated as apparent digestibility [(consumed − stool)/consumed]. These factors, contained in the data from this retrospective analysis, were used in this report. To correct for endogenous protein appearance in feces, calculated endogenous loss based on metabolic body size was subtracted from stool values, resulting in a calculation of TTTD = [consumed−(stool − endogenous protein)]/consumed, using the estimate of endogenous protein as previously described [12]. A TTTD for fat was similarly calculated using the method of Kendall et al. [12] to determine endogenous fat loss but was calculated using the current dietary intake and observed stool fat levels. Metabolizable energy (ME) was calculated using the methods outlined by the AAFCO, whereby digestible energy is measured and ME is then calculated using the correction factor for energy lost in urine for dogs (1.25 × g protein absorbed) and cats (0.86 × g protein absorbed) [5]. Hair was not removed from feces prior to analysis. Digestible carbohydrate was estimated using NFE due to the complexity of analysis required and because this is a retrospective study where no in vitro evaluations were possible. The specific factors used in this test exist because they were the complete set of measured values available for the evaluation of the stool quality and moisture content outcomes.

Stool firmness was evaluated using a grading system of 1 to 5 (Appendix A). Graders were trained and previously evaluated for both accuracy and precision. Feces that did not have solid form and were more than 75% liquid were given a grade of 1. Feces that were soft and mounded and approximately 50% solid and 50% liquid were given a grade of 2. Feces that had some cylindrical shape and were more than 75% formed and solid were given a grade of 3. Feces that were greater than 75% cylindrical and more than 50% firm were given a grade of 4. Feces that were cylindrically shaped and more than 80% firm were given a grade of 5. Feces were scored during the phase 2 collection period, with all firmness scores averaged to obtain a single stool firmness score per animal per test. Thus, stool firmness scores were approximated as a continuous rather than a categorical nominal variable.

The absolute amount of protein, fiber (crude), and fat that was not absorbed from the food was used to calculate ATTD. This was accomplished through multiplying the percent of protein, fiber, or fat in the feces by the average daily amount of feces collected, for example, grams stool protein = [stool % protein] × [grams feces collected]. The actual measurement of carbohydrate was not carried out in these studies. This was (among other factors) because carbohydrate measurement is not standard in digestibility studies because of the need for a complex analysis that is burdensome to complete for this number of studies.

Dietary fiber is known from published reports to reduce the digestibility of protein in the small intestine [13] and to increase fecal nitrogen as plant cell wall material and/or microbial biomass [14]. Fiber has also been shown to decrease fat digestibility [15] and increase endogenous fat loss [16]. Thus, the degree to which dietary fiber intake and stool fiber are associated with the reduced digestibility of protein and fat was determined from the current data by a mixed model.

### 2.4. Statistical Analyses

All statistical analyses were performed in JMP version 15 (SAS Institute, Cary, NC, USA). The analytic results for each diet were averaged to provide a single set of values representing that diet to account for diets that were studied in more than one digestibility test. Significance was determined at *p* ≤ 0.05. An analysis of variance (ANOVA) with Tukey’s post hoc multiple comparisons was used to assess the differences in diet nutrients, with the main effects being species (dog vs. cat) and product type (wet vs. dry) as well as an interaction term for the effect of species × product type. As there were a large number of data points, effect size (ES_NUTRIENT_) was estimated using Cohen’s D with a pooled, weighted standard deviation. This method estimates the size of between-group differences in standard deviation units, which helps avoid inflating the meaningfulness of highly significant but minimal differences. Effect size was classified according to Cohen’s importance where a negligible ES is that between −0.2 and 0.2, a small ES consists of absolute values between |0.2| and |0.5|, medium ES at |0.5| to |0.8| and a large effect size greater than |0.8| [17]. Nutrient intakes and stool bypass nutrients were scaled to metabolic bodyweight (g/kg BW^0.75^) to allow comparisons across species. Linear mixed modeling was used for all statistical tests where animal was the experimental unit in order to account for the inclusion of individual animals in more than one digestibility test; both animal and product type were random factors in these models. These included the tests for the effects of bypass nutrients on stool moisture and stool firmness, intake and age on digestibility, and age on intake, stool moisture and stool firmness. The coefficients and intercepts are reported in the tables. Coefficients of the mixed model parameters were used as de facto effect sizes in order to ascertain the relative contribution when these coefficients were determined to be significantly different from zero. Independent *t*-tests using the mean and SE of the mixed model parameter estimates (effect sizes) were performed to assess the relative magnitude of impact on TTTD and entrapment into the stool of protein and fat by fiber and for differences by species. 

## 3. Results

### 3.1. Food Compositions by Species and Product Type

Means, SE and confidence intervals of the compositions of the dog and cat foods tested in the digestibility studies are presented in Table 1. To establish the population of foods tested, the differences between product types and target species of these foods were assessed by ANOVA with Tukey’s post hoc multiple comparisons performed for all pairwise differences between the four groups (Table 1). The meaningfulness of these differences is provided as effect size (ES_NUTRIENT_) of [Dog–Cat] for both dry and wet foods, as well as [Dry–Wet] for dog and cat foods in Table 1, using Cohen’s weighted pooled SD. All examined nutrient parameters were statistically significant in the ANOVA overall model, and all nutrients were significantly influenced by both the main effects (species and product type) as well as the interactive effect of species × product type, except for crude fiber, which did not show an effect of the product type. 

When examining dog versus cat foods across both product types, the dog foods were lower in protein (ES_PROTEIN_: Dry(dog/cat) = −2.2; Wet(dog/cat) = −3.3) and fat (ES_FAT_: Dry(dog/cat) = −0.70; Wet(dog/cat) = −1.04), in agreement with AAFCO nutrient profiles for canine and feline foods. Additionally, NFE was higher in dog foods than cat foods (ES_NFE_: Dry(dog/cat) = +1.89; Wet(dog/cat) = +3.18), indicating that carbohydrate is used in dog foods to meet energy requirements at the expense of fat. There were no large differences in crude fiber content between dog and cat dry foods, although a small effect size showed that dry dog foods were marginally higher in crude fiber than dry cat foods (ES_FIBER_: Dry(dog/cat) = +0.34). Ash was marginally lower in both wet and dry dog foods compared to the equivalent cat food product type (ES_ASH_: Dry(dog/cat) = −0.22; Wet = −0.42), which is consistent with ash being associated with animal protein sources and the slightly increased minimums for many minerals for cats. Likely as a consequence of the reduced fat content of dog foods, gross energy (GE) was lower in these foods as well relative to the cat variants (ES_GE_: Dry(dog/cat) = −1.07; Wet(dog/cat) = −1.85). Dogs have a palatability preference for higher-moisture dry kibble, which is consistent with the observation that moisture levels in dry dog foods were increased relative to dry cat foods (ES_H2O_: Dry(dog/cat) = +0.40). Across species, the percent of total dietary calories stemming from protein, fat and carbohydrate largely paralleled the aforementioned observations described for the same nutrients when expressed on a dry matter basis. 

When the effect of dry versus wet product type was examined across species, protein was lower in dry than wet foods for cat variants (ES_PROTEIN_: Cat(dry/wet) = −1.0) but not meaningfully different between dry and wet dog foods. Fat was lower in dry than wet foods for both species (ES_FAT_: Dog(dry/wet) = −0.45; Cat(dry/wet) = −0.79), with the effect being stronger in cat foods. Dry cat food had increased NFE relative to cat wet variants (ES_NFE_: Cat(dry/wet) = +1.42) but was not meaningfully different by product type for dog foods. Similarly, fiber (ES_FIBER_: Cat(dry/wet) = −0.23) and ash (ES_ASH_: Cat(dry/wet) = −0.22) were reduced in dry vs. wet foods with a weak effect. In dog foods, fiber was marginally increased in dry foods relative to wet (ESFIBER: Dog(dry/wet) = +0.21), although there was no meaningful difference for ash by product type in dog foods. The GE was marginally higher in dry dog foods than wet, but this trend was reversed for cat foods (ES_GE_: Dog(dry/wet) = +0.29; Cat(dry/wet) = −0.50). Across product type, the percent of total dietary calories stemming from protein, fat and carbohydrate largely paralleled the aforementioned observations described for the same nutrients when expressed on a dry matter basis.

### 3.2. Estimated Loss of Endogenous Protein and Fat

The loss of endogenous integumental and metabolic protein through feces is used to calculate protein TTTD from ATTD. Endogenous protein loss for dogs and cats observed in this study was assessed by linear mixed modeling using protein intake per day (g/day) as the independent variable and stool protein (g/kg BW^0.75^/day) as the dependent variable, with animal and product type as random factors. At the extrapolated intercept of this regression, a theoretical point where no protein was consumed, the stool output of endogenous protein for dogs was estimated as (mean ± SE; [95% CI]; *p* for difference from zero for intercept) 461 ± 78; [−389, 1312] mg/kg BW^0.75^/day; *p* = 0.096. For cats, endogenous protein loss was estimated as 348 ± 15; [316, 380] mg/kg BW^0.75^/day; *p* < 0.01. Product type was included as a random factor in the mixed model but was not significant in either the dog or the cat model, indicating that the endogenous protein loss estimates reported here can be used for both dry and wet foods within the range of foods described in Table 1. 

Fat is also subject to endogenous losses from endogenous and metabolic sources. A calculation for endogenous fat loss carried out similarly to that for the loss of endogenous protein was carried out. At the intercept of the regression where theoretically no fat was consumed, the stool output of endogenous fat for dogs was estimated as (mean ± SE; [95% CI]; *p* for difference from zero for intercept) 272 ± 33; [−113, 657] mg/kg BW^0.75^/day; *p* = 0.07. For cats, endogenous fat was estimated as 193 ± 46; [−365, 752] mg/kg BW^0.75^/day; *p* = 0.14. Product type was included as a random factor in the mixed model but was not significant in either the dog or the cat model, indicating that the endogenous fat loss reported here can be used for both dry and wet foods similar to those described in Table 1. Although it had previously been reported that there were differences between puppies and adult dogs for endogenous fat loss [18], the current trial had insufficient numbers of animals to perform a direct comparison between adults and puppies (or kittens) less than 1 year old; in order to assess a quantitative effect of older versus younger animals, the dogs and cats, based on the available ages in this retrospective study, were subdivided into bins of greater than, or less than, 3 years of age. The breakpoint at 3 years of age was chosen to provide a group with a sufficient number which compared pets in the first quarter of their lifespan. In this subgroup analysis by binned age, dogs greater than 3 years of age (*n* = 5218) manifested an endogenous fat loss of 276 ± 29; [−30, 583] mg/kg BW^0.75^/day; *p* = 0.056. The value determined for dogs less than 3 years old (*n* = 571) was 282 ± 60; [−163, 728] mg/kg BW^0.75^/day; *p* = 0.090. These values for dogs older and younger than 3 years old were not deemed significantly different by an independent *t*-test (*p* = 0.95). Cats were also assessed for age differences with a breakpoint of 3 years old. Cats greater than 3 years of age (*n* = 4833) manifested an endogenous fat loss of 244 ± 54; [−238, 728] mg/kg BW^0.75^/day; *p* = 0.11. The value determined for cats less than 3 years old (*n* = 649) was 188 ± 42; [−304, 679] mg/kg BW^0.75^/day; *p* = 0.13. These values for cats older and younger than 3 years old were not deemed significantly different by an independent *t*-test (*p* = 0.63). A comparative value for endogenous loss for cats is not available in the literature. For the digestibility assessments reported hereafter, the values determined in the current study for endogenous protein and endogenous stool fat loss are used to convert ATTD to TTTD.

### 3.3. Nutrient Digestibilities by Species and Product Type

The means, SE and confidence intervals of the ATTD and TTTD for the foods are presented in Table 2. The significance of the effect of species and product type as well as their interaction on ATTD and TTTD was assessed for significance (Table 2). The magnitude of the differences in ATTD and TTTD by species [Dog–Cat] and product type [Dry–Wet] were assessed by effect size using Cohen’s D with weighted pooled SD (Table 2) in the same manner as for food compositions. 

Considering the effect of species, dogs had decreased protein TTTD compared to cats (ES_PROTEIN_: Dry(dog/cat) = −0.40; Wet(dog/cat) = −0.78), a small to medium species effect which was numerically 50% greater in wet than dry foods. In contrast, fat TTTD was increased in dog versus cats for both dry and wet foods, a medium-sized species effect (ES_FAT_: Dry(dog/cat) = +0.53; Wet(dog/cat) = +0.52). Carbohydrate NFE ATTD, a proxy for starch digestibility, was also higher in dogs than cats for both dry and wet foods, and the effect size was numerically more than twice as great in wet than dry foods (ES_NFE_: Dry(dog/cat) = +0.33; Wet(dog/cat) = +0.79). Both total dry matter and fiber ATTD were increased in dogs versus cats for wet foods only, with a small effect magnitude (ESDM: Wet(dog/cat) = +0.34, ES_FIBER_: Wet(dog/cat) = +0.37). For the effect of product type on digestibility, dog dry foods had increased protein TTTD relative to wet variants (ES_PROTEIN_: = +0.36), but there was no difference by product type for cat foods. Both species exhibited increased fat TTTD in dry versus wet foods (ES_FAT_: Dog(dry/wet) = +0.53; Cat(dry/wet) = +0.60). The NFE ATTD was increased in dry vs. wet cat foods (ESNFE: Cat(dry/wet) = +0.47) and not different by product type for dog foods. While fiber ATTD was not different by product type for cats, it was reduced in dry dog foods relative to their wet variants with a medium effect size (ESFIBER: Dog(dry/wet) = −0.42). Dry matter ATTD was increased in dry versus wet cat foods with a small effect size, but there was no effect of product type in dog foods (ESDM: Cat(dry/wet) = +0.24). There were significant interactive effects of species × product type for all measured digestibilities by mixed modeling (Table 2). Given the influence of sex hormones on fluid balance and gastrointestinal function, these data were assessed for differences by sex in intact young animals. There were no differences between young intact male and female animals (dogs or cats) in TTTD for fat or protein, nor were there any differences by sex in stool firmness or moisture. Although they would not be due to the influence of sex hormones, the data were also assessed for differences between males and females in adult/neutered animals. There were no differences by sex in spayed/neutered adult dogs for TTTD for fat or protein, nor were there any differences by sex in stool firmness or moisture. Intriguingly, in spayed/neutered cats, there was a statistically significant, albeit a small effect size, effect of sex on stool firmness and moisture such that female cats had approximately 9% higher subjective stool firmness (*p* < 0.01) and ~5% lower (relative) moisture (*p* < 0.01). Since this effect of sex in adult cats was not present in dogs, it could not be due to the hormonal influence of sex hormones and was of a small effect size, so it was not included in the statistical models for cats. In part, this was to enable cross-species comparisons in the data by utilizing the same model for both species; the models would be incomparable across species if sex had been included as a factor for spayed/neutered adult cats only. 

### 3.4. Impact of Undigested (Bypass) Protein, Fat and Fiber on Stool Moisture and Stool Firmness

These evaluations used stool moisture and the subjective measurement of stool firmness to define stool quality. The variable of pet age influences intake and digestibility (both of which influence bypass nutrients) and has a direct effect on stool quality. The relationship of the changing dietary concentrations of the macronutrients with age influencing stool quality is shown in Figure 1. 

First, the direct effects of measured bypass macronutrients on stool moisture and subjective stool firmness were assessed. The association between percent moisture in feces and stool bypass nutrients was examined (Table 3). There was no significant effect of product type as a random factor and this was removed from the model. Stool protein was associated with higher stool moisture for both species. Stool fat was positively associated with stool moisture in dogs but not cats, while fiber was negatively associated with stool moisture in both dogs and cats. The interaction of protein × fat decreased stool moisture in dogs only, while both the interaction of protein × fiber and fat × fiber increased stool moisture in cats only. The three-way interaction of protein × fat × fiber was associated with reduced stool moisture in cats and had no association in dogs.

The relative association of stool moisture as a response to bypass nutrients found in feces was assessed for protein and fiber, which were the only model effects significantly different from zero in both species. The parameter estimates for the main effect of stool protein on stool moisture was not different between species. However, the magnitude of the parameter estimate for stool fiber was significantly smaller in dogs compared to cats (*p* < 0.01), meaning that stool bypass fiber had a greater effect in decreasing stool moisture in cats vs. dogs. 

Although stool moisture levels are a primary driver of observed stool firmness, additional factors such as water-holding capacity and the nature of the constituents of stool may lead to lack of concordance between the objective measure of stool moisture with the subjective measure of stool firmness score. Therefore, the impact of nutrients that had bypassed digestion and absorption on subjective stool firmness was assessed using mixed modeling (Table 4). The cat model failed to converge when product type was included as a random factor, although when product type was tested alone for cats, its effect by mixed modeling was not significant (a parameter estimate for the product type effect on stool score in cats; mean = −0.0096, 95% CI = −0.025, 0.006; *p* = 0.24). Product type was not significant as a random factor in the dog model, and thus, product type was excluded from both species’ models. In parallel with its effect to increase stool moisture in dogs and cats, this stool protein was negatively associated with stool firmness in both species (dog *p* = 0.01; cat *p* < 0.01). Stool fat was positively associated with stool firmness in cats (*p* < 0.01), but there was no significant association in dogs. Aligned with expectations based on the effects of stool moisture, stool fiber was positively associated with stool firmness in both dogs and cats (*p* < 0.01). No significant interaction of protein × fat affecting stool firmness arose from the models. There was a significant interaction between protein × fiber to decrease stool firmness in dogs (*p* = 0.01), but no such effect in cats. In contrast, cats manifested a significant interaction of fat × fiber to decrease the stool firmness score (*p* = 0.01), but there was no effect of this interaction in dogs. Finally, the three-way interaction of protein × fat × fiber weakly increased stool firmness in dogs (*p* = 0.04) but not in cats. 

The relative association of stool firmness with bypass nutrients found in feces was assessed for species difference in dogs versus cats in the same way as performed for stool moisture; as was the case for moisture, only protein and fiber main effects were significantly different from zero in both dog and cat. Cat stool firmness was more strongly impacted by dietary protein (*p* < 0.01) and fiber (*p* < 0.01) than was dog stool firmness. The magnitude of dog parameter estimates for the effects of stool protein and fiber was 85% to 63% lower than those observed for cats, implying that protein and fiber were both stronger determinants of stool firmness in cats than they were in dogs. 

### 3.5. Fiber Entrapment of Dietary Protein and Fat

Fiber intake and fecal fiber influence protein and fat digestibility and fecal amounts (Appendix A). A negative association was found by linear mixed modeling between dietary crude fiber intake scaled to metabolic body weight (g crude fiber/kg^0.75^/day) and protein TTTD for both dogs and cats (*p* < 0.01 both species). The magnitude of the cat parameter estimate for the effect of fiber intake to decrease protein TTTD was 29% larger than the dog estimate, and this difference was significant (*p* = 0.01). Similarly, dietary crude fiber intake decreased fat TTTD in both dogs and cats (*p* < 0.01 for both). The magnitude of the cat parameter estimate for the effect of fiber intake to decrease fat TTTD was 248% larger than the dog estimate, and this difference was significant (*p* < 0.01). Next, an assessment of the magnitude of the parameter estimates for the dietary fiber decrement in protein TTTD versus fat TTTD was performed within a species; this indicated the degree to which dietary fiber intake decreased protein TTTD relative to fat TTTD in dogs or cats and showed that, for dogs, the estimate for the effect of fiber to decrease protein TTTD was 247% higher than for fat TTTD (*p* < 0.01 for difference). In contrast, for cats, the estimate for protein TTTD was only 29% higher than the estimate for fat TTTD (*p* = 0.05 for difference). Taken together, the relative capacity for dietary fiber to decrease protein TTTD to a greater degree than it decreased fat TTTD was higher for dogs than for cats; dietary fiber intake in cats was much more equitable in its effect of decreasing protein TTTD and fat TTTD.

It would be expected that the above results describing the effect of fiber to decrease protein and fat TTTD would also manifest as an association between stool fiber and stool protein and fat. Thus, we performed an analysis by linear mixed modeling to determine the degree to which stool fiber was associated with stool protein and fat (Appendix A). All stool bypass nutrients were scaled to metabolic body weight for these analyses and expressed as g/kg^0.75^/day. Stool crude fiber was positively associated with stool protein for both dogs and cats (*p* < 0.01 for both species). The magnitude of the cat parameter estimate for effect was 42% larger than the dog estimate; this difference was significant (*p* < 0.01). Stool fiber also was positively associated with stool fat in both dogs and cats (*p* < 0.01). The magnitude of the cat parameter estimate for the association of stool fiber with stool fat was 51% larger than the dog estimate, and this difference was significant (*p* < 0.01). Finally, the magnitude of the parameter estimates for stool fiber to increase stool levels of protein versus fat within a species was determined; this indicated the degree to which stool fiber entrapped protein versus fat in dogs or cats. In this analysis, for dogs, the estimate of stool protein entrapment was 512% higher than the estimate for stool fiber entrapping stool fat (*p* < 0.01 for difference). In contrast, for cats, the estimate of stool protein entrapment was 475% higher than the estimate for stool fiber entrapping stool fat (*p* < 0.01 for difference); thus, the strength of association of stool protein with stool fiber relative to the association of stool fat with stool fiber was somewhat similar in cats and dogs.

### 3.6. Determinants of Digestive Bypass 

Next, given the impact of undigested nutrients on stool firmness found in this study, age and intake were assessed as determinants of digestibility, as well as the effect of age on intake. The relationships examined are directional such that age can affect nutrient intake, and both age as well as nutrient intake can affect digestibility, but the causal directionality of these relationships is not reversible, e.g., digestibility and intake do not cause changes in chronological age (Figure 1). It should be noted that in the digestibility studies reported here, all animals were fed to maintain bodyweight, and thus, nutrient intakes are de facto proxy assessments of the provision of the nutrients in food sufficient to keep body weight constant from start to finish of the test. With increasing age, animals typically exhibit decreasing activity and other changes in metabolism. Therefore, it is to be expected that the amount of food provided to these subjects is reduced according to the equation for caloric provision to maintain body weight (activity factor × 70 × kg BW^0.75^). 

First, the effect of age on the nutrient digestibility of protein, fat, NFE and fiber was assessed, modeling age as both a linear and a quadratic term in a mixed model in order to gain insights into the degree to which nutrients bypass digestion to potentially impact stool quality with advancing age (Table 5). Age had a significant positive linear relationship with protein TTTD in cats (*p* < 0.01) but not dogs. With advanced age, there was a decrease in protein TTTD in cats (*p* < 0.01) with a negative quadratic term; again, there was no relationship in dogs. The same was true for fat TTTD in cats, such that there was a positive linear relationship (*p* < 0.01) and a negative quadratic relationship (*p* < 0.01). There was no linear or quadratic effect of age on fat TTTD in dogs. Cat NFE ATTD had no linear relationship with age, but a trend towards increased NFE ATTD with advancing age (quadratic term *p* = 0.06) existed. There was no effect of age (linear or quadratic) on dog NFE ATTD. Both dogs and cats also had a negative linear association between age and fiber ATTD (*p* < 0.01), although there was no effect of age as a linear term in dogs. Further, there was a positive effect of age as the quadratic on fiber ATTD for both dogs (*p* < 0.01) and cats (*p =* 0.01). 

Next, the effect of age on the dietary nutrient intake of protein, fat, NFE and fiber was assessed, again modeling age as both a linear and a quadratic term in a mixed model in order to gain insights into factors determining the amount of ingested nutrients available to bypass digestion (Table 6). Age had a significant negative linear relationship with protein intake in both dogs (*p* < 0.01) and cats (*p* < 0.01). With a sufficiently advanced age, there was an increase in protein intake for cats (*p* < 0.01) but not dogs. In cats, the same was true for fat intake, such that there was a negative linear relationship (*p* < 0.01) and a positive quadratic relationship (*p* < 0.01). In dogs, there was no relationship between age and fat intake. The same pattern was present for cats in the association between age and NFE intake; both a negative linear age term (*p* < 0.01) and a positive quadratic age term (*p* < 0.01) were significant. For dogs, the only association between age and NFE was a negative linear age term (*p =* 0.01). Cats exhibited a positive linear age term (*p* < 0.01) and a negative quadratic age term (*p* < 0.01). In dogs, there was no association between age and fiber intake. For metabolizable energy (ME), cats had a significant negative linear association (*p* < 0.01) but a positive quadratic association (*p* < 0.01) with age. Dogs manifested only a negative quadratic association between age and ME intake (*p =* 0.01).

Given that age impacted nutrient intake, the degree to which protein, fat, NFE or fiber intake may have impacted that nutrient’s respective digestibility was ascertained (Table 7). Both species exhibited the same pattern of significance and directionality in the linear and quadratic associations between the intake of a given nutrient and the digestibility of that nutrient. Thus, in both cats and dogs, there was a negative linear relationship between protein intake and protein TTTD, fat intake and fat TTTD, and fiber intake and fiber ATTD (*p* < 0.01 for all linear terms in both species). Further, for both species, there was a positive relationship for the quadratic relationship between a nutrient’s intake and its digestibility (*p* < 0.01 for all quadratic terms in both species). There was a positive linear relationship between NFE intake and NFE TTTD in both dogs and cats (both species *p* < 0.01). Further, there was a negative relationship between the quadratic NFE intake term and NFE ATTD in cats (*p* < 0.01) but no such relationship for dogs. 

Given the impact of age on nutrient digestibility and intake, the association of age with indices of stool quality, e.g., stool moisture and stool firmness, was assessed (Table 8). Age was not associated with stool moisture as a linear term for either dogs or cats. Age as a quadratic term was not associated with stool moisture in dogs but was negatively associated in cats (*p* = 0.04). Intriguingly, cats and dogs manifested the opposite age-related patterns of stool firmness such that cat age had a significant negative linear (*p* = 0.02) and positive quadratic association with stool firmness, while dog age had a positive linear (*p* < 0.01) and negative quadratic (*p* < 0.01) association with stool firmness.

## 4. Discussion 

This study is the most comprehensive investigation reported to date on the characteristics of dog and cat companion animal foods, the influence of bypass nutrients on stool firmness, and the impact of age and intake on macronutrient digestibility. 

As expected, the proximate analysis of foods in the current study confirms expectations of increased protein and fat in cat foods compared to dog foods and dogs’ preference for higher moisture content in dry foods. Additionally, the observations confirm the logical consequence of the aforementioned nutrient profiles: in dogs, decreased protein and fat is supplanted by increased NFE. In these data, dietary ash accompanies dietary protein, although with the advent of increased vegetable source proteins (e.g., corn gluten meal, rice protein concentrate and lentils) and meat protein isolates, this correlation may no longer hold in diets that rely heavily on vegetable protein or meat protein isolates. Similarly, dog foods have increased fiber compared to cat foods. Perhaps this fiber is present as an accompaniment to the increased plant-derived NFE. The intestinal microbiomes of both domesticated dogs [18,19,20] and cats [21,22] possess the capacity to metabolize starch and fiber, which results in the production of microbial metabolites. These data also confirm that cat foods are more ketogenic in nature, certainly due to their increased employment of fat as a source of calories at the expense of NFE, even though cat foods, on balance, do not fully rise to a level classically considered nutritionally ketogenic [23]. 

A reassessment of the loss of endogenous protein and fat in dogs showed that these data largely agree with previously published values, differing by 17% for protein [11] and 5–7% for fat [13,24]. Endogenous nitrogen loss (metabolic stool nitrogen) in dogs has been estimated to be 63 ± 2.9 mg/kg BW^0.75^/day [11], which translates to an intercept of 394 mg/kg BW^0.75^/day for protein in dogs. The estimate from the current study was 462 mg/kg BW^0.75^/day. The higher estimate in this study may reflect an actual higher production of both sloughed cells and digestive enzymes or may be a result of the extrapolation approach taken in the current study versus that taken in the previous report where the actual intake of protein was zero [11]. There was seemingly only one report on stool nitrogen loss in cats; this yielded a calculated value for stool protein loss of 168 mg protein/kg BW^0.75^ per day using hair-free stool [25]. In the current report, protein in cat stool yielded a value for endogenous protein loss of 384 mg protein/kg BW^0.75^ per day. However, this was determined using stool that contained hair the cats had consumed during self-grooming and was thus expected to overestimate stool protein loss. Taken together, this suggests that the true difference in endogenous nitrogen loss between dogs and cats may be even greater than estimated above.

The value for endogenous fat loss has been reported to be 155 mg/kg BW/day in adult beagle dogs [24], the predominant type of dog enrolled in the digestibility tests reported here. Given the average body weight of 12.7 kg for dogs in the current report and using the endogenous fat loss value from Marx et al. [24], this yields a value for endogenous fat loss of 292 mg/kg BW^0.75^/day, a value that differs by 7% from the value reported here. In another report, a value of 138 mg/kg BW/day endogenous fat loss was reported in adult beagles and 262 mg/kg BW/day in 5-month-old puppies [18]; these values equated to 260 mg/kg BW^0.75^/day for adult dogs (12.6 kg from [18]) and 396 mg/kg BW^0.75^/day for puppies (5.2 kg from [18]). The value for adults differs by less than 5% from the current study. Data from the current study found no significant differences in endogenous fat loss between dogs (or cats) that were younger versus older than 3 years, which indicates that by the time of maturation, the loss of endogenous fat had stabilized. The current work also extended findings for endogenous protein and fat loss to cats. In contrast to dogs, these data did not show that endogenous fat loss was significantly different than zero for cats. Since there are no previously peer-reviewed published values available to convert protein ATTD to TTTD for cats, these data may find use in personalizing nutrition for adult cats. 

The domestication of dogs has led to increased genetic capacity for the metabolism of both starch and fat [25,26], and while cats are obligate carnivores [27], they change protein oxidation with intake [28], even though they maintain a level of gluconeogenesis from proteogenic amino acids independent of intake [29,30]. On this basis, it is teleologically tempting to ascribe an evolutionary aspect to the observation here that dogs exhibited increased digestive efficiency toward starch and fat than did cats, while cats manifested increased digestive efficiency toward protein. The patterns in digestibility held for both dry and wet product types; since product type held no sway, it could be that these are inherent species differences. It should be noted that cats appear to be fully capable of digesting starch with high efficiency, even though they may produce different glucose and insulin responsiveness compared to dogs [31].

The primary goal of this study was to assess the degree to which protein, fat and fiber can impact digestibility and stool quality metrics either individually or by association. The data show that both dogs and cats exhibited the same pattern of influence of protein and fiber on stool moisture levels and stool firmness. Protein digestive bypass increased the objective metric of stool moisture and concurrently decreased subjective stool firmness. The importance of examining how nutrients undergoing digestive bypass interact to impact stool quality is evident for the case of fiber; while the main effect of fiber considered individually was to decrease stool moisture and increase stool firmness in both dogs and cats, when a significant interaction of fiber with either protein or fat was present, the effect was the opposite. This translated into differences in stool firmness, such that fiber increased stool firmness as a main effect, but protein × fiber and fat × fiber decreased stool moisture in cats and protein × fiber decreased stool firmness in dogs, while fat × fiber decreased stool firmness in cats. The presence of significant interactions between fiber and other nutrients suggests that it is important for pet food manufacturers to consider the total diet composition, and the potential presence of nutrients in an excess of digestive capacity, when predicting the effect of fiber on stool firmness, and this may help explain the differing effects of fiber observed in practice. In the present study, given that dogs presented lower protein digestive efficiency and cats lower fat digestive efficiency (vide infra), it may be that these differences in digestibility then translated into meaningful differences in stool quality once those bypass nutrients interacted with dietary fiber.

The magnitude of effect of bypass nutrients to impact stool quality was markedly increased for cats versus dogs. In every instance where a bypass nutrient was significantly associated with stool moisture or stool firmness in both cats and dogs, the effect size was reduced in dogs compared to cats. This indicates the degree to which greater attention must be directed toward the formulation of cat foods in order to impact stool quality; dogs are relatively imperturbable to digestibility-induced changes in stool quality relative to cats.

Dietary fiber intake was associated with decrements in protein and fat TTTD in both species. A trend in pet food manufacturing is to supplement pet foods with fiber sources for purposes such as the promotion of weight loss and satiety and for hairball control. Fiber is known to entrap other dietary components (e.g., protein and fat) during the digestion process, resulting in decreased digestibility [32,33], and thus might be expected to participate in interactions that influence stool firmness. The current data add insight by showing that the decrement in protein TTTD by fiber was greater than it was for decrement in fat TTTD in both dogs and cats, although the effect was much greater in dogs than cats. Also, potentially relevant to the commercial manufacturing of companion animal foods is the observation of dietary fiber decreasing protein TTTD to a greater degree than fat TTTD. In contrast, dietary fiber intake in cats led to a more equitable decrement in protein TTTD versus fat TTTD. A parallel assessment showed that stool fiber was entrapped with stool protein or fat in both dogs and cats and that stool fiber entrapped stool protein about four times more than it did stool fat; this magnitude of protein versus fat trapping by fiber was remarkably similar across species, unlike the effect of dietary fiber on protein versus fat TTTD. The observation that stool protein and fat were more entrapped by co-occurring fiber in cats than in dogs may reflect the differences in the microbes of their gastrointestinal tracts, where the dog microbiome has a greater capacity to degrade cellulosic materials, thereby decreasing the amount of fiber available to entrap bypass protein and fat. The different methods used to estimate the impact of fiber on protein and fat in stool (dietary fiber vs. protein or fat TTTD, stool fiber vs. stool protein or fat) did not yield entirely consistent effect sizes, and it is difficult to pinpoint a reason for the different estimates. The estimates differed most in cats (protein vs. fat TTTD 29% compared to protein vs. fat fecal level 475%) compared to dogs (protein vs. fat TTTD 247% compared to protein vs. fat fecal level 512%). Nevertheless, these results suggest that dietary fiber decreases protein digestibility to a greater extent than fat, particularly in dogs, and stool fiber traps protein to a greater extent than fat. Future food formulations may seek an optimal “sweet spot” for the relative proportions of fiber and protein for dogs versus cats. 

Analogous to the degree in which cats were more susceptible to the perturbance of stool quality with the appearance of nutrients in feces, this species also exhibited more associations between age and nutrient digestibility. Whereas dogs did not manifest significant interactions between age and protein, fat or NFE digestibility, cats manifested significant interactions between age and protein, fat and NFE digestibility. In contrast, both species showed a similar pattern of the relationship between age and fiber digestibility. Overall, age appears to more strongly impact cat digestive efficiency than it does dog. While the pets in these trials were fed to maintain body weight, changes in the provision of nutrients and energy with age may indicate the degree to which nutritional requirements change across their lifespan. Consistent with the presumption that, with increasing age, pets reduce their activity levels and experience other changes in metabolism, in the earlier stages of aging, cats needed less protein, fat, NFE and energy (ME) to maintain body weight, while this was true for protein and NFE only for dogs. At a sufficiently advanced age (the quadratic term), this pattern was reversed for cats such that cats required more protein, fat, carbohydrate and ME to maintain body weight. Dogs showed no such positive relationship between the requirement of protein, fat, carbohydrate or ME at an advanced age but rather a decrement in the amount of ME needed. If we assume that ME intakes are similar to maintenance requirements in a weight-stable population, this finding agrees with data from 319 French dogs, collected over a period of more than 10 years. The authors concluded that maintenance energy requirements are a function of age and slightly decrease over a lifespan [34]. In contrast, however, are the findings of a meta-analysis of 29 studies in which the authors concluded that there were no differences in the maintenance energy requirements comparing young adult dogs to old dogs; it should be noted that these authors converted chronological age into ‘biological’ age by adjusting chronological age according to breed size, which may explain the incongruency of their results with other publications that used untransformed chronological age [35]. Perhaps the unifying conclusion that can be drawn from all of these studies, including the findings presented in this paper, is that any changes in ME intake with age are likely to be relatively small, with the intakes of older animals being less than 10% lower than the intakes of younger animals. It is unlikely that these age-related changes in energy intake could confound the observed changes in digestibility with age. 

Across all ages, increasing macronutrient intake was associated with decreasing protein, fat and fiber digestibility but increasing carbohydrate digestibility. However, in nearly every case, the pattern reversed at the highest levels of intakes (the quadratic term). Since this study employed particularly relevant study specific metrics for endogenous protein and fat loss to arrive at TTTD, it is unlikely that the decrease in these nutrients’ digestibility was due to the amount lost from endogenous sources. There is no clear explanation apparent for the reversal of the directionality of digestibility at the highest intake levels. It could be that conditioning of the gut microbiota towards the nutrients to which it was most exposed led to an increased capacity of microbiota to degrade digestive bypass nutrients to compounds not measured in stool proximate analyses due to their resorption into the host or the presence in feces as molecules not detected in the assays. It may also be a compensatory response by the dogs and cats to adapt to new environmental conditions of increased nutrient provision.

Turning to previously published reports for context, one prior study assessed differences in protein and fat digestion between old (10 years) and adult dogs (2.5 years). Although there was no interactive effect of age × soluble fiber level across the three levels of soluble fiber employed, a post hoc analysis indicated that old dogs but not adult dogs fed a diet containing 1.2% fermentable fiber had reduced protein and fat digestibility when compared to a diet containing ostensibly no soluble fiber [36]. In the current study, there was a significant effect of age on dog and cat fiber digestibility and on cat protein and fat digestibility. Fiber digestibility was minimized in mid-adult life (8 years in cats and 11 years in dogs), while protein digestibility and fat digestibility in dogs were maximized at that time (fat at 8 years and protein at 11 years). As the total tract digestibility of fiber is significantly influenced by gut microbiota, this suggests that the mid-adult-life microbiota may be more attuned to saccrolytic metabolism. The current study did not differentiate between fermentable and non-fermentable fiber but rather used crude fiber as the metric for fiber intake and stool levels. Crude fiber analysis preferentially responds to insoluble versus soluble fiber, and insoluble fibers on balance tend to be non-fermentable (e.g., cellulose, lignin). Although it is not possible to draw a straight parallel, it would appear that the results presented here are to some degree discordant with previous observations [30] as we observed that crude fiber intake decreased both protein and fat digestibility across a wide swath of ages (1 year to 16 years old) in both cats and dogs. There may be differences between purified cellulose and the sources of detectable crude fiber utilized in the foods comprising test materials for the current report compared to the purified cellulose added to cat diets that did not significantly decrease protein or fat digestibility [37]. In contrast, the foods utilized herein derived detectable crude fiber from both native plant forms as well as added cellulose. Additionally, animal-based ingredients may provide a source of crude fiber that is capable of bypassing digestion; this form of crude fiber may be particularly relevant to cats [38]. Other studies in dogs showed no effect of age on digestibility [39,40,41]. Body size of the dog also seems to affect digestibility, as greater digestibility has been observed in large versus smaller breeds [42,43]. In the current study, beagles comprised 88% of dog subjects, and thus, a subgroup analysis by body size or breed was not possible. A previous report indicated that among cats, the apparent digestibility of most macronutrients was highest in mature cats, intermediate in young adult cats and lowest in old cats regardless of dietary energy content, and there was a significant negative linear coefficient for the association between age and fat digestibility regardless of dietary energy content [44]. The current report is consistent with those findings; the data herein show a positive linear association in cats with age and either protein or fat digestibility but a negative association at a sufficiently advanced age. Another study also found that older cats had decreased fat digestibility compared with younger cats [6]. It has been reported that mature and old cats fed a high-energy diet were able to digest crude fiber nominally better compared to young cats, which was consistent with the current data showing a negative quadratic association between age and crude fiber ATTD [44]. Overall, these previous studies employed relatively small numbers of animals but were planned interventions, whereas the current study is a retrospective analysis but leverages a large number of animals over extended periods. Taken together, however, these studies suggest lower crude fiber digestibility with increasing age in cats and possibly in dogs. In the current study, dogs exhibited neither a linear nor a quadratic effect of age on true protein TTTD. However, cats exhibited a positive age effect on protein TTTD and a negative quadratic effect with advanced age. It is important to acknowledge though that the magnitude of this effect is small from a nutritional effect size standpoint. Whereas a 1-year-old cat is predicted to have a protein TTTD of 90.9% according to the parameters observed in Table 5, that value reaches a maximum of 94% at 11 years old and then declines to 93% at 17 years old. Thus, the total impact of age observed in the current study would be limited to a range of approximately 3% across a lifetime. While increased protein intake in aged animals may help mitigate age-related sarcopenia, decreased protein digestibility would not appear to be a reason for concern in companion animal nutrition. To overcome a perceived loss of digestive capacity in older animals, an increasing trend in the pet food industry has been to supplement the diets of elderly animals with additional protein. However, an overabundance of protein has been associated with poor stool firmness [42,45] and potentially other adverse health effects [46,47]. For example, protein spillover past the digestive and absorptive region in the small intestine makes this macronutrient available to the gut microbiome, which carries out proteolysis and putrefaction to produce amino acid metabolites known to detrimentally impact health during disease and aging [47]. While the current data indicate that increasing protein intake may improve its digestibility, merely increasing the protein content of food for older pets without a clear understanding of digestibility may increase the cost of the food without additional health benefits and expose dogs and cats to potential detriments. Therefore, future work aimed at identifying the optimal level of protein intake across life stages for the benefit of pet health, pet owner financial burden and decreased output nitrogen burden with accompanying environmental benefits [48] would be valuable. 

There were significant limitations in this study. First, as a retrospective analysis, there was no opportunity for increasing the variables available for evaluation. Most importantly, variables further defining the foods were not available (e.g., soluble and insoluble fiber, the source of protein and fat, the amount of resistant starch). Finally, confounding was present in the data; for example spay/neuter status was almost entirely confounded with age, making the independent evaluation of sex and spay/neuter status impossible. It is worth noting that there is also a confounding of sex hormone differences between species that may account for the species differences, but without hormonal analysis, this cannot be evaluated.

## 5. Conclusions

Understanding changes in macronutrient digestibility that may occur as pets age, as well as the impact of stool bypass nutrients on stool firmness, is important for developing diets that optimize health throughout their lifespan. The current report indicates an opportunity to improve dietary protein and fat availability in cats as they reach an advanced age and proposes that dogs may not require the same consideration. However, in both cats and dogs, fiber ‘digestibility’ (i.e., fermentation) decreases with age throughout adulthood, leading to the possibility that increasing the proportion of readily fermentable fiber in the diet of adult dogs and cats who are not yet senior in age may offset age-related changes. In addition, the results of the present study allow an assessment of the impact on stool firmness of food formulation changes with known digestibility outcomes. These results should help drive future investigations into pet foods that provide the appropriate balance of nutrients to maximize digestibility while minimizing waste.

## Figures and Tables

**Figure 1 animals-14-02778-f001:**
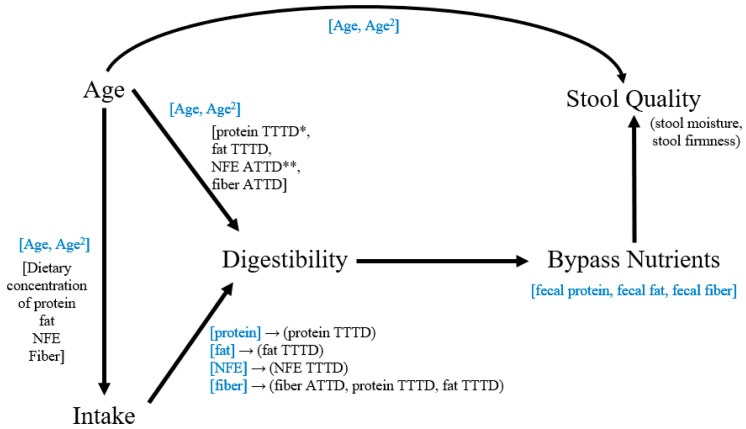
The control of stool quality as influenced by age, dietary intake, subsequent digestibility and bypass nutrients. * True total tract digestibility. ** Apparent total tract digestibility. Factors that are in blue type significantly influence the factors that follow after the arrows or are inside the parenthesis in black type.

**Table 1 animals-14-02778-t001:** Characterization of macronutrient food content by intended species and product type effect on macronutrient food content expressed as means, std errors and Cohen’s D effect size using pooled, weighted standard deviation.

Diet Analyte *	Effect Size				Product and Species								
	Canine Dry−Wet	Feline Dry−Wet	Dry Canine −Feline	Wet Canine− Feline	Product Type	Species Name	Overall Model *p*	Main and Interaction Effects Significance ^§^	Least Squares Mean	Std Error	Lower 95%	Upper 95%	*n*
Protein Crude (%)	0.09	−1.00	−2.20	−3.30	Dry	Canine	<1 × 10^−10^	S, T, S×T	24.1 ^a^	0.183	23.7	24.5	744
						Feline			35.1 ^b^	0.205	34.7	35.5	593
					Wet	Canine			23.6 ^c^	0.335	23.0	24.3	222
						Feline			40.1 ^a^	0.277	39.5	40.6	326
Fat Crude (%)	−0.45	−0.79	−0.70	−1.04	Dry	Canine	<1 × 10^−10^	S, T, S×T	15.9 ^d^	0.172	15.6	16.3	744
						Feline			19.2 ^b^	0.193	18.8	19.6	593
					Wet	Canine			18.0 ^c^	0.315	17.4	18.7	222
						Feline			22.9 ^a^	0.260	22.4	23.4	326
Fiber Crude (%)	0.21	−0.23	0.34	−0.09	Dry	Canine	<1 × 10^−8^	S, S×T	5.1 ^a^	0.144	4.8	5.4	744
						Feline			3.7 ^c^	0.162	3.4	4.0	593
					Wet	Canine			4.3 ^b,c^	0.264	3.8	4.8	222
						Feline			4.6 ^b,c^	0.218	4.2	5.1	326
NFE (%) *	0.12	1.42	1.89	3.18	Dry	Canine	<1 × 10^−10^	S, T, S×T	49.3 ^a^	0.259	48.8	49.8	744
						Feline			36.0 ^b^	0.290	35.4	36.6	593
					Wet	Canine			48.5 ^a^	0.474	47.6	49.4	222
						Feline			26.0 ^c^	0.392	25.2	26.8	326
Ash (%)	−0.02	−0.22	−0.22	−0.42	Dry	Canine	<1 × 10^−10^	S, T, S×T	5.6 ^c^	0.068	5.4	5.7	744
						Feline			6.0 ^b^	0.076	5.8	6.1	593
					Wet	Canine			5.6 ^b,c^	0.124	5.3	5.8	222
						Feline			6.4 ^a^	0.102	6.2	6.6	326
Gross Energy (kcal)	0.29	−0.50	−1.07	−1.85	Dry	Canine	<1 × 10^−10^	S, T, S×T	5085.5 ^c^	11.411	5063.1	5107.9	744
						Feline			5419.0 ^b^	12.781	5393.9	5444.1	593
					Wet	Canine			4996.4 ^d^	20.889	4955.4	5037.3	222
						Feline			5573.5 ^a^	17.238	5539.7	5607.3	326
Moisture (%)	−21.07	−22.13	0.40	−0.66	Dry	Canine	<1 × 10^−10^	S, T, S×T	8.1 ^c^	0.116	7.9	8.4	744
						Feline			6.9 ^d^	0.130	6.6	7.1	593
					Wet	Canine			74.7 ^b^	0.212	74.3	75.1	222
						Feline			76.8 ^a^	0.175	76.4	77.1	326
Digestible Energy (kcal)	7.83	8.68	−0.86	−0.01	Dry	Canine	<1 × 10^−10^	S, T, S×T	4048.8 ^b^	13.850	4021.7	4076.0	744
						Feline			4374.0 ^a^	15.514	4343.6	4404.4	593
					Wet	Canine			1092.1 ^c^	25.355	1042.3	1141.8	222
						Feline			1095.4 ^c^	20.923	1054.3	1136.4	326
Metabolizable Energy (kcal)	7.50	8.37	−0.85	0.01	Dry	Canine	<1 × 10^−10^	S, T, S×T	3814.6 ^b^	13.601	3787.9	3841.3	744
						Feline			4130.1 ^b^	15.235	4100.2	4160.0	593
					Wet	Canine			1031.1 ^c^	24.899	982.3	1079.9	222
						Feline			1026.2 ^c^	20.547	985.9	1066.5	326
Protein Calories (%)	0.26	−0.67	−1.67	−2.59	Dry	Canine	<1 × 10^−10^	S, T, S×T	21.9 ^c^	0.208	21.5	22.4	744
						Feline			31.4 ^b^	0.233	30.9	31.9	593
					Wet	Canine			20.5 ^d^	0.380	19.8	21.2	222
						Feline			35.2 ^a^	0.314	34.6	35.8	326
Fat Calories (%)	−0.34	−0.73	−0.62	−1.01	Dry	Canine	<1 × 10^−10^	S, T, S×T	33.7 ^d^	0.271	33.1	34.2	744
						Feline			38.2 ^b^	0.303	37.7	38.8	593
					Wet	Canine			36.2 ^c^	0.496	35.2	37.2	222
						Feline			43.7 ^a^	0.409	42.9	44.5	326
NFE ^Ω^ Calories (%)	0.15	1.27	1.93	3.05	Dry	Canine	<1 × 10^−10^	S, T, S×T	44.4 ^a^	0.266	43.9	44.9	744
						Feline			30.4 ^b^	0.298	29.8	30.9	593
					Wet	Canine			43.3 ^a^	0.488	42.3	44.3	222
						Feline			21.1 ^c^	0.402	20.3	21.9	326

^§^: species main effect (S), product type main effect (T), interactive effect (S×T). Abbreviations only shown if parameter estimate from model is significant at *p* < 0.05; * All analytes reported as percent total mass on a dry matter basis, except gross energy and metabolizable energy, which are on an as-fed basis; a,b,c,d: means with different letters within a diet analyte row are different, *p* < 0.05; **^Ω^** nitrogen-free extract and estimate of carbohydrate.

**Table 2 animals-14-02778-t002:** Characterization of nutrient digestibility of foods by species and product type expressed as means, standard errors and Cohen’s D effect size using pooled, weighted standard deviation.

Digestibility	Effect Size				Product and Species			Descriptive Stats				
	Canine Dry−Wet	Feline Dry−Wet	Dry Canine −Feline	Wet Canine− Feline	Product Type	Species Name	Main and Interaction Effects Significance ^§^	Least Square Means (%)	Std Err Mean	Lower 95% Mean	Upper 95% Mean	*n*
Ash ATTD ^α^	−0.12	−0.34	−0.09	−0.32	Dry	Canine	S, T, ST	46.45 ^d^	0.179	46.10	46.80	4525
						Feline	S, T, ST	47.63 ^b^	0.214	47.21	48.05	3728
					Wet	Canine	S, T, ST	47.96 ^c^	0.363	47.25	48.67	1264
						Feline	S, T, ST	52.09 ^a^	0.315	51.47	52.71	1754
NFE * ATTD ^α^	−0.04	0.47	0.33	0.79	Dry	Canine	S, T, ST	89.02 ^a^	0.081	88.87	89.18	4525
						Feline	S, T, ST	87.19 ^b^	0.095	87.01	87.38	3728
					Wet	Canine	S, T, ST	89.25 ^a^	0.117	89.02	89.48	1264
						Feline	S, T, ST	84.14 ^c^	0.185	83.78	84.50	1754
Dry Matter ATTD ^α^	−0.07	0.24	0.01	0.34	Dry	Canine	S, T, ST	83.71 ^b^	0.096	83.52	83.90	4525
						Feline	S, T, ST	83.67 ^a,b^	0.089	83.49	83.85	3728
					Wet	Canine	S, T, ST	84.14 ^a^	0.140	83.87	84.42	1264
						Feline	S, T, ST	82.35 ^c^	0.129	82.09	82.60	1754
Gross Energy ATTD ^α^	0.12	0.43	9.1 × 10^−4^	0.29	Dry	Canine	S, T, ST	86.65 ^a^	0.087	86.48	86.82	4525
						Feline	S, T, ST	86.64 ^a^	0.080	86.49	86.80	3728
					Wet	Canine	S, T, ST	85.97 ^b^	0.132	85.72	86.23	1264
						Feline	S, T, ST	84.51 ^c^	0.126	84.26	84.76	1754
Fiber ATTD ^α^	−0.42	−0.06	0.09	0.37	Dry	Canine	S, T, ST	32.67 ^c^	0.298	32.09	33.25	4525
						Feline	S, T, ST	30.73 ^b^	0.416	29.91	31.54	3728
					Wet	Canine	S, T, ST	41.26 ^a^	0.608	40.06	42.45	1264
						Feline	S, T, ST	32.37 ^a^	0.622	31.15	33.59	1754
Fat TTTD ^Ω^	0.53	0.60	0.53	0.52	Dry	Canine	S, T, ST	99.69 a	0.032	99.63	99.75	4525
						Feline	S, T, ST	98.33 ^b^	0.049	98.23	98.42	3728
					Wet	Canine	S, T, ST	98.35 ^b^	0.099	98.16	98.55	1264
						Feline	S, T, ST	96.09 ^c^	0.118	95.86	96.32	1754
Protein TTTD ^Ω^	0.36	−0.05	−0.40	−0.78	Dry	Canine	S, T, ST	90.79 ^b^	0.068	90.66	90.92	4525
						Feline	S, T, ST	92.62 ^a^	0.075	92.47	92.77	3728
					Wet	Canine	S, T, ST	89.09 ^c^	0.143	88.81	89.37	1264
						Feline	S, T, ST	92.84 ^a^	0.111	92.62	93.06	1754

^§^: species main effect (S), product type main effect (T), interactive effect (S×T). Abbreviations only shown if parameter estimate from model is significant at *p* < 0.05; a,b,c,d: means with different letters within a digestibility row are different, *p* < 0.05; ^α^ apparent total tract digestibility; ^Ω^ true total tract digestibility; * nitrogen-free extract and estimate of carbohydrate.

**Table 3 animals-14-02778-t003:** The impact of undigested protein, fat, and fiber on stool moisture in dogs and cats.

Parameter	Canine					Feline				
	Prob > |t|	Estimate	Std Error	95% Lower	95% Upper	Prob > |t|	Estimate	Std Error	95% Lower	95% Upper
Intercept		63.095	1.285	54.440	71.750		62.399	0.596	61.079	63.719
Protein	<0.001	6.420	0.412	5.612	7.228	<0.001	7.401	0.588	6.248	8.555
Fat	<0.001	10.373	1.616	7.206	13.541	0.516	−1.216	1.872	−4.886	2.454
Fiber	<0.001	−2.539	0.346	−3.216	−1.861	<0.001	−11.448	1.031	−13.470	−9.427
Protein × Fat	<0.001	−7.895	1.116	−10.082	−5.707	0.293	−1.597	1.518	−4.573	1.378
Protein × Fiber	0.206	0.263	0.208	−0.145	0.671	<0.001	4.361	0.940	2.518	6.205
Fat × Fiber	0.068	1.753	0.959	−0.128	3.633	<0.001	20.112	3.479	13.292	26.931
Protein × Fat × Fiber	0.822	0.106	0.471	−0.817	1.029	<0.001	−10.938	2.750	−16.329	−5.546

**Table 4 animals-14-02778-t004:** The impact of undigested protein, fat and fiber on subjective stool firmness.

Parameter	Canine					Feline				
	Prob > |t|	Estimate	Std Error	95% Lower	95% Upper	Prob > |t|	Estimate	Std Error	95% Lower	95% Upper
Intercept		4.304	0.055	4.195	4.412		4.349	0.050	4.250	4.447
Protein	0.011	−0.099	0.039	−0.176	−0.023	<0.001	−0.651	0.055	−0.759	−0.544
Fat	0.129	0.235	0.155	−0.069	0.539	<0.001	0.660	0.176	0.315	1.004
Fiber	<0.001	0.155	0.033	0.091	0.219	<0.001	0.435	0.097	0.244	0.626
Protein × Fat	0.083	−0.183	0.106	−0.390	0.024	0.171	−0.197	0.144	−0.480	0.085
Protein × Fiber	0.005	−0.055	0.020	−0.093	−0.016	0.406	−0.074	0.089	−0.248	0.100
Fat × Fiber	0.538	−0.056	0.090	−0.233	0.122	0.014	−0.810	0.328	−1.453	−0.166
Protein × Fat × Fiber	0.044	0.090	0.045	0.002	0.177	0.088	0.443	0.260	−0.066	0.952

**Table 5 animals-14-02778-t005:** The statistical relationship of age with protein fat, NFE * and fiber digestibility.

Digestibility	Parameter	Canine					Feline				
		Prob > |t|	Estimate	Std Error	95% Lower	95% Upper	Prob > |t|	Estimate	Std Error	95% Lower	95% Upper
Protein TTTD ^Ω^	Intercept		89.862	0.378	89.121	90.603		90.300	0.386	89.542	91.057
	Age	0.797	0.028	0.109	−0.186	0.242	<0.0001	0.674	0.118	0.442	0.906
	Age × Age	0.633	0.004	0.007	−0.011	0.018	0.0002	−0.032	0.009	−0.049	−0.015
Fat TTTD ^Ω^	Intercept		99.134	0.233	98.676	99.592		96.064	0.319	95.438	96.689
	Age	0.339	−0.062	0.065	−0.189	0.065	<0.0001	0.511	0.098	0.318	0.704
	Age × Age	0.105	0.007	0.004	−0.001	0.016	<0.0001	−0.031	0.007	−0.045	−0.017
NFE ATTD ^α^	Intercept		89.137	0.366	88.419	89.854		86.671	0.511	85.668	87.673
	Age	0.698	−0.041	0.106	−0.249	0.167	0.275	−0.173	0.159	−0.484	0.138
	Age × Age	0.588	0.004	0.007	−0.010	0.018	0.060	0.021	0.011	−0.001	0.044
Fiber ATTD ^α^	Intercept		34.226	1.882	30.534	37.917		45.512	2.108	41.380	49.645
	Age	0.0001	−1.892	0.493	−2.858	−0.927	<0.0001	−2.861	0.641	−4.118	−1.604
	Age × Age	<0.0001	0.134	0.033	0.070	0.199	0.005	0.132	0.047	0.040	0.223

^α^ Apparent total tract digestibility; ^Ω^ true total tract digestibility; * nitrogen-free extract and estimate of carbohydrate.

**Table 6 animals-14-02778-t006:** The linear and quadratic effect of age on nutrient intake in dogs and cats.

Nutrient (G/KG BW^0.75^)	Parameter	Canine					Feline				
		Prob > |t|	Estimate	Std Error	95% Lower	95% Upper	Prob > |t|	Estimate	Std Error	95% Lower	95% Upper
Protein	Intercept		8.735	0.215	8.314	9.156		8.106	0.151	7.810	8.402
	Age	<0.01	−0.183	0.060	−0.300	−0.066	<0.0001	−0.466	0.047	−0.558	−0.375
	Age × Age	0.327	0.004	0.004	−0.004	0.012	<0.0001	0.021	0.003	0.015	0.028
Fat	Intercept		5.693	0.158	5.384	6.002		4.453	0.097	4.263	4.643
	Age	0.306	−0.045	0.044	−0.131	0.041	<0.0001	−0.220	0.030	−0.279	−0.161
	Age × Age	0.126	−0.005	0.003	−0.010	0.001	<0.0001	0.008	0.002	0.004	0.012
NFE **	Intercept		17.613	0.393	16.843	18.383		7.227	0.177	6.880	7.574
	Age	<0.01	−0.286	0.106	−0.494	−0.077	<0.0001	−0.349	0.055	−0.456	−0.241
	Age × Age	0.653	−0.003	0.007	−0.017	0.011	<0.0001	0.013	0.004	0.006	0.021
Fiber	Intercept		0.078	0.007	0.065	0.091		0.019	0.002	0.015	0.023
	Age	0.588	−0.001	0.002	−0.005	0.003	<0.0001	0.004	0.001	0.003	0.005
	Age × Age	0.129	2.0 × 10^−4^	1.3 × 10^−4^	−5.8 × 10^−5^	0.000	<0.0001	−2.8 × 10^−4^	4.7 × 10^−5^	−3.7 × 10^−4^	−1.8 × 10^−4^
Metabolizable Energy *	Intercept		145.069	2.830	139.519	150.619		98.027	1.839	94.420	101.634
	Age	0.056	−1.404	0.734	−2.844	0.035	<0.0001	−5.205	0.569	−6.320	−4.090
	Age × Age	0.028	−0.108	0.049	−0.204	−0.011	<0.0001	0.220	0.041	0.139	0.300

* (kcal/KG BW^0.75^); ** nitrogen-free extract and estimate of carbohydrate.

**Table 7 animals-14-02778-t007:** The effect of protein, fat, NFE and fiber intake on the subsequent digestibility of protein, fat, NFE ** and fiber.

Parameter *		Canine					Feline				
		Prob > |t|	Estimate	Std Error	95% Lower	95% Upper	Prob > |t|	Estimate	Std Error	95% Lower	95% Upper
Protein TTTD ^Ω^	Intercept		99.234	0.453	98.346	100.12		99.205	0.330	98.557	99.852
	Protein	<0.001	−1.866	0.102	−2.066	−1.67	<0.001	−1.351	0.071	−1.491	−1.211
	Protein × Protein	<0.001	0.083	0.006	0.072	0.09	<0.001	0.051	0.004	0.043	0.058
Fat TTTD ^Ω^	Intercept		101.413	0.237	100.947	101.88		99.942	0.225	99.500	100.384
	Fat	<0.001	−0.665	0.075	−0.813	−0.52	<0.001	−0.777	0.077	−0.927	−0.626
	Fat × Fat	<0.001	0.037	0.006	0.026	0.05	<0.001	0.039	0.006	0.027	0.051
NFE ATTD ^α^	Intercept		85.155	0.605	83.969	86.34		78.906	0.421	78.081	79.731
	NFE Carbohydrate	<0.001	0.267	0.071	0.128	0.41	<0.001	2.164	0.124	1.920	2.407
	NFE × NFE	0.542	−0.001	0.002	−0.005	0.003	<0.001	−0.122	0.009	−0.140	−0.104
Fiber ATTD ^α^	Intercept		34.802	0.995	32.846	36.76		40.640	0.963	38.750	42.530
	Fiber	<0.001	−101.257	7.515	−115.990	−86.52	<0.001	−264.003	24.007	−311.067	−216.939
	Fiber × Fiber	<0.001	131.401	18.094	95.930	166.87	<0.001	782.157	171.554	445.837	1118.477

* Nutrient intake (G/KG BW^0.75^); ** nitrogen-free extract and estimate of carbohydrate; ^α^ apparent total tract digestibility; ^Ω^ true total tract digestibility.

**Table 8 animals-14-02778-t008:** The effect of age on stool moisture and firmness.

Metric	Parameter	Canine					Feline				
		Prob > |t|	Estimate	Std Error	95% Lower	95% Upper	Prob > |t|	Estimate	Std Error	95% Lower	95% Upper
Fecal Moisture	Intercept		69.294	0.470	68.371	70.216		65.797	0.606	64.609	66.986
	Age	0.786	−0.036	0.133	−0.296	0.224	0.367	0.167	0.185	−0.195	0.529
	Age × Age	0.614	−0.005	0.009	−0.022	0.013	0.039	−0.028	0.013	−0.054	−0.001
Stool Firmness Score	Intercept		3.960	0.052	3.859	4.062		4.057	0.060	3.939	4.174
	Age	<0.001	0.113	0.013	0.088	0.138	0.015	−0.043	0.018	−0.077	−0.008
	Age × Age	<0.001	−0.008	0.001	−0.010	−0.006	<0.001	0.006	0.001	0.003	0.008

## Data Availability

The original contributions presented in the study are included in the article/Appendix A, and further inquiries can be directed to the corresponding author.

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
