# Peer review of "Nutrient Digestive Bypass: Determinants and Associations with Stool Quality in Cats and Dogs"

_animals, 2024, doi:10.3390/ani14192778_

Round 1

Reviewer 1 Report

Comments and Suggestions for Authors

Stool Bypass Nutrients Impact Stool Firmness in Dogs and Cats

**please see PDF of paper with additional comments and suggested amended/expansions/clarifications**

Overall comments

Thank for you the opportunity to review this detailed, comprehensive, interesting, and relevant paper exploring the impact of bypass nutrients on stool quality in dogs and cats. This is an expansive retrospective study that considers a large data set and explores key nutritional outcomes from a portfolio of studies. Outcomes and conclusions are of relevance to our understanding of companion animal food preparation and formulation, as well as our understanding of digestive processes and outcomes.

This is a current area of interest and relevant in terms of companion animal nutrition.

Overall, I have little to really comment on and critically, the work is competent and detailed without excessive extrapolation – the paper is well written, considered and constructed. I felt confident with the data presentation and interpretation which appears robust and thorough. The dataset is enormous because of the retrospective analysis – this makes the work useful in nutrition work.

There is clear background, relevance and justification to the work and the outcomes are informative.

The premise of the work is clear, and the undertaking and analysis appears relevant and well undertaken (I confess NOT to be an expert in some of the modelling undertaken).

Ethical review is noted and is clear.

Overall structure, flow and language use is clear and supports readability but there could be enhanced paragraph use to aid flow in areas (although I appreciate there might be space limitations) – some paragraphs are long and contain several key pieces of information that can get lost/lose focus.

Overall, I feel the paper is expansive and detailed with copious information and content that is relevant to the wider field.

Keywords – relevant and appropriate for this paper. Perhaps add companion animal and faecal quality?

Figures and tables are fine and suitable for the work BUT I would like to see some additional formatting and clarity of them all to aid readability and highlighting of key information – ensure that text can always been read in full and remove additional ‘ghost’ gridlines. Perhaps consider more colour formatting (red and green are not ideal choices for accessibility) to aid clarity.

References

I have not exhaustively gone through these, but all appear fine – present and correct although I have not proofread or cross referenced to check validity of use. Check all are in correct format.

Author Response

Overall comments

Thank for you the opportunity to review this detailed, comprehensive, interesting, and relevant paper exploring the impact of bypass nutrients on stool quality in dogs and cats. This is an expansive retrospective study that considers a large data set and explores key nutritional outcomes from a portfolio of studies. Outcomes and conclusions are of relevance to our understanding of companion animal food preparation and formulation, as well as our understanding of digestive processes and outcomes.

This is a current area of interest and relevant in terms of companion animal nutrition.

 We are grateful for the opportunity to address the comments from yourself and the reviewers. These insights and questions were helpful in revising the manuscript to be stronger and more focused. We hope that we have sufficiently addressed the comments

Overall, I have little to really comment on and critically, the work is competent and detailed without excessive extrapolation – the paper is well written, considered and constructed. I felt confident with the data presentation and interpretation which appears robust and thorough. The dataset is enormous because of the retrospective analysis – this makes the work useful in nutrition work

Thank you

There is clear background, relevance and justification to the work and the outcomes are informative.

 Thank you

The premise of the work is clear, and the undertaking and analysis appears relevant and well undertaken (I confess NOT to be an expert in some of the modelling undertaken).

Ethical review is noted and is clear.

 Thank you

Overall structure, flow and language use is clear and supports readability but there could be enhanced paragraph use to aid flow in areas (although I appreciate there might be space limitations) – some paragraphs are long and contain several key pieces of information that can get lost/lose focus.

We have made changes to the abstract, introduction and discussion to better focus the text, improve readability and flow, and emphasize the key pieces of information.

Overall, I feel the paper is expansive and detailed with copious information and content that is relevant to the wider field.

  Thank you

Keywords – relevant and appropriate for this paper. Perhaps add companion animal and faecal quality?

  We have added key words as requested

Figures and tables are fine and suitable for the work BUT I would like to see some additional formatting and clarity of them all to aid readability and highlighting of key information – ensure that text can always been read in full and remove additional ‘ghost’ gridlines. Perhaps consider more colour formatting (red and green are not ideal choices for accessibility) to aid clarity.

  The color was removed as it was unclear.  The figures and tables were redone for clarity..

References

I have not exhaustively gone through these, but all appear fine – present and correct although I have not proofread or cross referenced to check validity of use. Check all are in correct format.

We have reworked the references for the correct format

Reviewer 2 Report

Comments and Suggestions for Authors

Thank you for submitting your work for review. The manuscript attests to a large amount of work and data analysis. Unfortunately, the length of the manuscript does detract from its readability as it must be well over 10,000 words. I therefore recommend that the authors refine the manuscript to be more succinctly written with a tighter focus on the primary goal of the study with all additional information being supplementary. Alternatively, the manuscript could be split into more than one publication. 

I have provided an annotated PDF with my specific comments adn queries which need to be addressed.

My general comments for each section are below.

Title: A simpler title would most likely engage more readers. For example: Nutrient digestive bypass: determinants and associations with stool quality in cats and dogs.

Abstract: The abstract is more than twice the length indicated in the authors' instructions which state "The abstract should be a total of about 200 words maximum." The results in the abstract lack precision.

Introduction: The introduction covers sufficient information to provide context for the study. In places, supporting literature/references need to be cited. 

Methods: Clarification is required in several places as indicated in the annotated PDF. Some of the analyses need to be better justified. Much more detail is needed in the statistical analyses section (this can be provided as supplementary information). Each of the GLMs used should be detailed including the equation for the model. Justification for the factors included is also required. For example, it is not clear why product type would be included in a model investigating nutrient bypass on stool quality especially if a preceding analysis has already examined the association of food type with nutrient bypass. Some of the models appear to be over-specified. Sufficient detail needs to be provided so that the same analysis can be repeated on a similar set of data.

Why were sex and whether or not the animal was desexed not considered/corrected for in the analyses? This requires an explanation. Given the effects of female sex hormones on intestinal motility and fluid balance, I would have thought this was important to factor in. 

Was any analysis of the outliers performed? Did the outlier have anything in common? For example, if they were a dog were they a non-Beagel? 

Given the association of intestinal health with the microbiota there was an opportunity here to also investigate the microbial presence in each of the stool samples. Was this done?

For any given food type with a species were there enough animals tested at different ages to carry out a repeated measures ANOVA to verify any age effects detected with the GLM?

Results: The results need to be more focused on the main topic of the article. There are a lot of supplementary analyses presented in the main article which detract from the key results that align with the focus of the article. The focus should be on the models that determine the factors that influence nutrient bypass and the models that determine the factors associated with stool quality.

Tables need to be formatted according to the authors' guidelines and not be a snip of the spreadsheet. The authors' guidelines state " Authors should use the Table option of Microsoft Word to create tables." Given the number of tables and how large some of them are, I would recommend moving the larger more complex tables to supplementary information as the key results are written in the text.

Discussion: The discussion could be more concisely written if the re-iteration of results and speculation was removed. A section on the limitations of the study needs to be added.

References: The list well exceeds the maximum of 30 as specified in the authors' instructions. The formatting of the references lacks consistency.

Supplementary Tables - The tables need informative titles and explanatory captions. The word "Removel" needs to be spelled correctly. Abbreviations need to be defined and units added where appropriate.

Author Response

Thank you for submitting your work for review. The manuscript attests to a large amount of work and data analysis. Unfortunately, the length of the manuscript does detract from its readability as it must be well over 10,000 words. I therefore recommend that the authors refine the manuscript to be more succinctly written with a tighter focus on the primary goal of the study with all additional information being supplementary. Alternatively, the manuscript could be split into more than one publication. 

We are very appreciative of the opportunity to present these data for consideration, and are grateful for your insightful and detailed comments.

I have provided an annotated PDF with my specific comments adn queries which need to be addressed.

Thank you for this level of detail, it was exceptionally helpful. We have tried to address each of the points and feel that your review has strengthened the report. 

My general comments for each section are below.

Title: A simpler title would most likely engage more readers. For example: Nutrient digestive bypass: determinants and associations with stool quality in cats and dogs.

This was rewritten and the title simplified.

Abstract: The abstract is more than twice the length indicated in the authors' instructions which state "The abstract should be a total of about 200 words maximum." The results in the abstract lack precision.

This was rewritten and focused as requested.

Introduction: The introduction covers sufficient information to provide context for the study. In places, supporting literature/references need to be cited. 

The references have been reworked.  However, we do not think that there is a need for references when the information is so general as to be included in multiple textbooks.

Methods: Clarification is required in several places as indicated in the annotated PDF. Some of the analyses need to be better justified. Much more detail is needed in the statistical analyses section (this can be provided as supplementary information). Each of the GLMs used should be detailed including the equation for the model.

We have read through the annotated PDF and made many of the suggested changes.  We do not think that repeating the actual detailed equations is valuable as the factors (i.e., the intercept and coefficients) are included in the tables.

Justification for the factors included is also required. For example, it is not clear why product type would be included in a model investigating nutrient bypass on stool quality especially if a preceding analysis has already examined the association of food type with nutrient bypass. Some of the models appear to be over-specified. Sufficient detail needs to be provided so that the same analysis can be repeated on a similar set of data.

We have included this justification in the methods.  The models were used because it was a retrospective study where these are the complete set of the factors available.   The models are directly represented in the tables so it is quite possible for other researchers to evaluate both the model and the coefficients from other data sets.

Why were sex and whether or not the animal was desexed not considered/corrected for in the analyses? This requires an explanation. Given the effects of female sex hormones on intestinal motility and fluid balance, I would have thought this was important to factor in. 

We did assess the effect of sex as well as spay/neuter status as part of our assessment of factors which influence digestibility and stool firmness.  However, this assessment is fraught, since spay/neuter status is inherently confounded with age. Thus, there are no adult or older animals that are intact, and there are no puppies or kittens which were spayed/neutered. So the effect of spay/neuter status is inherently confounded with age. This is the reason we performed the analysis of the effect of age as a quadratic in our models so we could determine whether there were inflections in the effect of age at advanced age vs very young ages. To address the reviewer's point about hormonal influence of estrogens on fluid balance and GI function, we can look for differences by sex in unspayed/un-neutered intact young animals. Our data show that there were no differences between young intact male and female animals (dogs or cats) in TTTD for fat or protein, nor were there any differences by sex in stool firmness or moisture in these young intact dogs and cats. We also assessed any sex differences in adult, spayed/neutered animals even though these differences would not be due to the hormonal influences described by the reviewer. There were no differences by sex in spayed/neutered adult dogs for TTTD for fat or protein, nor were there any differences by sex in stool firmness or moisture. Interestingly, in spayed/neutered cats there was a statistically significant, albeit small effect size, effect of sex on stool firmness and moisture such that female cats had approximately 9% higher subjective stool firmness and ~5% lower (relative moisture). Since this effect of sex seen in adult cats was not present in dogs, it could not be due to the influence of sex hormones and was of a small effect size, thus  we elected to not include the effect of sex in our statistical models for cats. One reason for this was to enable cross-species comparisons in the data by utilizing the same model for both species. If sex had been included as a factor for cats only, and only in spayed/neutered adults, then the models would have been incomparable across species. We have revised the draft manuscript to report these findings in the hopes that they may be of use to spur future research into possible effects of sex in adult, spayed/neutered cats. 

Was any analysis of the outliers performed? Did the outlier have anything in common? For example, if they were a dog were they a non-Beagel? 

The outliers were examined for patterns that might provide insight into commonalities, but there were no apparent patterns. There were no species, breed, sex, age, or weight patterns that distinguished the outliers from the final data set. It would appear that the outliers were a result of analytical variation and error in analysis of the fecal samples, since if the analytical error had occurred in the food then it would have impacted the entire panel of dogs on that particular digest test. So, we conclude there were no patterns that unified the outliers.

Given the association of intestinal health with the microbiota there was an opportunity here to also investigate the microbial presence in each of the stool samples. Was this done?

Due to the retrospective nature of this historical data, samples were not available for microbiome analysis. We do appreciate the reviewer's comment about the importance of the microbiome on total tract digestibility and stool firmness, and we are performing followup interventional studies which will assess microbiome determinants of digestibility and stool quality.

For any given food type with a species were there enough animals tested at different ages to carry out a repeated measures ANOVA to verify any age effects detected with the GLM?

We interpret food type here to mean wet or dry food.  We interpret the request to ask whether there were enough animals at different ages to examine the effect of varying age on nutrient digestibility, nutrient intake, stool moisture and stool firmness within wet food only and similarly within dry food only.  While we have explored the effect of age itself on the range of endpoints included in the paper (for example, Table 2 summarizes nutrient digestibility of foods by species and food type.  Tables 5 and 6 summarize the effect of age on digestibility and nutrient intake, respectively, while Table 8 summarizes the effect of age on stool moisture and firmness), we have not explored the effect of age on nutrient digestibility, nutrient intake, stool moisture and stool firmness within a particular food type such as wet or dry.  It is true that we observed differences in protein, fat, NFE and fiber digestibility by food type though the patterns were not always consistent between cats and dogs (Section 3.3). We appreciate the reviewer’s suggestion while recognizing that we may need to balance the size and scope of the present manuscript to maintain the paper’s focus.  That said, a deeper exploration of the effect of age within a given food type may be potentially interesting to explore in future work. 

Results: The results need to be more focused on the main topic of the article. There are a lot of supplementary analyses presented in the main article which detract from the key results that align with the focus of the article. The focus should be on the models that determine the factors that influence nutrient bypass and the models that determine the factors associated with stool quality.

Thank you for this feedback.  We agree that the results section could be streamlined and have removed material not related to the core points of the paper from the results section to improve readability.

Tables need to be formatted according to the authors' guidelines and not be a snip of the spreadsheet. The authors' guidelines state " Authors should use the Table option of Microsoft Word to create tables." Given the number of tables and how large some of them are, I would recommend moving the larger more complex tables to supplementary information as the key results are written in the text.

We have reworked and reformatted the tables.  However,  we think it important to allow the viewer to have this information in the main paper and we have left them in the manuscript.

Discussion: The discussion could be more concisely written if the re-iteration of results and speculation was removed. A section on the limitations of the study needs to be added.

We agree that the discussion section could be written more concisely and have removed material not related to the core points of the paper from this section to improve readability.   We have added a section on limitations as requested.

References: The list well exceeds the maximum of 30 as specified in the authors' instructions. The formatting of the references lacks consistency.

We have corrected the formatting.  We have surpassed the suggested maximum but feel the citations are required to adequately place these results with the previously published literature.

Supplementary Tables - The tables need informative titles and explanatory captions. The word "Removel" needs to be spelled correctly. Abbreviations need to be defined and units added where appropriate.

 We have corrected the tables as requested.

Reviewer 3 Report

Comments and Suggestions for Authors

Dear Authors,

Below you can find suggestions to improve your manuscript.

Title – is unclear and should be revised to reflect lines

8-45. Abstract – should be sorted according to Animals' instructions. Omit words: objective, methods, etc. It is unusual to use them in an abstract. Scientific English should be used in writing to more clearly reflect the aim and key findings of the manuscript.
Line 67 – sentence is not finished (by..)
Lines 77-83 – are not relevant to the present research and should be omitted.
Lines 95-98 – the aim is too broad and does not really match the scope of the paper. It should be omitted or narrowed down to reflect the scope of the paper.
Lines 156-158 – please clarify which fiber you are referring to? – crude or TDF, as these terms are not interchangeable, and use them consistently throughout the manuscript.
Line 212 – The chapter on results is too extensive and should be more concise. The tables in the chapter should be adapted to follow Animals' instructions. And the results are presented in a clear and concise manner that is standard scientific practice. The description of the table should present the results and be self-explanatory. Example: Table 1 – letters reflecting significance should be superscripted, no explanation of significance level is given, etc.
Line 522 – Discussion – it is unusual and confusing to use brackets too often and refer to previous conclusions without context. Should be revised to be clearer.
Lines 537-541 - The results are neither relevant nor applicable as the diets used are common feline diets which are a far from a ketogenic diet. Should be omitted.
603-607 it is presumptuous to claim lower digestible capacity as it could be that the diets had lower digestibility for certain nutrients.
In lines 675-687 (and elsewhere) in the discussion, the authors use high level of presumptions, without providing references. These parts of the discussion should be briefly and clearly rewritten, as they neither reflect the results nor relate the results to the relevant literature.
Finally, this study has many drawbacks – such as analyzing a large amount of data with a high degree of variability. This should be discussed in detail.

Comments on the Quality of English Language

Minor editing of English language required

Author Response

Dear Authors,

Below you can find suggestions to improve your manuscript.

Title – is unclear and should be revised to reflect lines

The title has been changed as requested.

8-45. Abstract – should be sorted according to Animals' instructions. Omit words: objective, methods, etc. It is unusual to use them in an abstract. Scientific English should be used in writing to more clearly reflect the aim and key findings of the manuscript.

This was completed as suggested.

Line 67 – sentence is not finished (by..)

This was changed as suggested.

Lines 77-83 – are not relevant to the present research and should be omitted.

This was extensively rewritten.  It was important to clarify that to understand the influence of the bypass nutrients the factors of endogenous supply, digestibility and age need to be considered.
Lines 95-98 – the aim is too broad and does not really match the scope of the paper. It should be omitted or narrowed down to reflect the scope of the paper.

This was rewritten to respond to this concern and enhance clarity.

Lines 156-158 – please clarify which fiber you are referring to? – crude or TDF, as these terms are not interchangeable, and use them consistently throughout the manuscript.

This was rewritten, we always use crude fiber and attempted to make this distinction clear.

Line 212 – The chapter on results is too extensive and should be more concise. The tables in the chapter should be adapted to follow Animals' instructions. And the results are presented in a clear and concise manner that is standard scientific practice. The description of the table should present the results and be self-explanatory. Example: Table 1 – letters reflecting significance should be superscripted, no explanation of significance level is given, etc.

This was rewritten, we have changed the tables to unify the presentation, added levels of significance.
Line 522 – Discussion – it is unusual and confusing to use brackets too often and refer to previous conclusions without context. Should be revised to be clearer.

This was rewritten as requested and reduced the bracket use.
Lines 537-541 - The results are neither relevant nor applicable as the diets used are common feline diets which are a far from a ketogenic diet. Should be omitted.

This was rewritten and the ketogenic discussion and references removed as requested.
603-607 it is presumptuous to claim lower digestible capacity as it could be that the diets had lower digestibility for certain nutrients.

This was rewritten as requested
In lines 675-687 (and elsewhere) in the discussion, the authors use high level of presumptions, without providing references. These parts of the discussion should be briefly and clearly rewritten, as they neither reflect the results nor relate the results to the relevant literature.

This was rewritten as requested.
Finally, this study has many drawbacks – such as analyzing a large amount of data with a high degree of variability. This should be discussed in detail.

This was rewritten and a discussion of limitations included.

Comments on the Quality of English Language

Minor editing of English language required

We have rewritten much of the paper with a desire to improve the language and clarity of the paper.

Reviewer 4 Report

Comments and Suggestions for Authors

The manuscript describes a meta-analysis investigating bypass nutrients impacts on stool firmness. The manuscript is generally well written. However, there are some shortcomings (results that do not address the objective) and misplaced text that need to be addressed.

Title: shorten to: Bypass Nutrients Impact Stool Firmness in Dogs and Cats

Lines 8-44: The abstract should be rewritten to provide information about the study results without repeating the objective and methods.

Lines 66-67: complete the sentence

Line 92: replace "study" with "meta analysis"

Lines 214-266: this does not address the objective; remove

Lines 290-307: this belongs in Discussion

Lines 312-317: delete

Lines 319-341: this does not address the objective; remove

Lines 343-344: delete "The overall strategy for parsing age, intake and digestibility as determinants of stool quality, defined here as stool moisture and subjective stool firmness, is shown in Figure 1."

Lines 362-391: belongs in Discussion

Lines 403-444: belongs in Discussion

Lines 445-458: belongs in Discussion

The Discussion and Conclusions should be rewritten to focus on the Objective. Furthermore, results of the study should be summarized in the Conclusions.

Author Response

The manuscript describes a meta-analysis investigating bypass nutrients impacts on stool firmness. The manuscript is generally well written. However, there are some shortcomings (results that do not address the objective) and misplaced text that need to be addressed.

Thank you for your careful review, we have done an extensive rewrite of the manuscript and included specific responses below to address what you have written.

Title: shorten to: Bypass Nutrients Impact Stool Firmness in Dogs and Cats

We have shortened the title as requested.

Lines 8-44: The abstract should be rewritten to provide information about the study results without repeating the objective and methods.

There was much desired in abstract changes.  We have changed it in a way that seems most likely to be acceptable to all reviewers.

Lines 66-67: complete the sentence

We have changed this and in that change completed the sentence.

Line 92: replace "study" with "meta analysis"

We have changed this as requested.

Lines 214-266: this does not address the objective; remove

These have been rewritten.  We do feel that a discussion of the actual foods consumed is an important component of understanding the results. We have completely removed the statements about ketogenic calculations and references to ketogenesis.

Lines 290-307: this belongs in Discussion

We have rewritten this; we do feel that what is now written in results is appropriate.  What is currently included is actual description of results rather than an interpretation, that component is now included as discussion.

Lines 312-317: delete

There was a disagreement between reviewers on this passage.  We have rewritten it which we believe answers your concern.  Basically, the data are valuable in that to understand the effect of the bypassed nutrients this is foundational.

Lines 319-341: this does not address the objective; remove

There was the disagreement we mentioned in the earlier lines on this deletion between reviewers.  As in the earlier lines – we have rewritten it which we believe answers your concern.  Basically, the data are valuable in that to understand the effect of the bypassed nutrients this understanding is foundational.

Lines 343-344: delete "The overall strategy for parsing age, intake and digestibility as determinants of stool quality, defined here as stool moisture and subjective stool firmness, is shown in Figure 1."

We have changed this section as requested by a number of reviewers.  Although we have not deleted it as it seemed essential because of the value of being able to look at the illustration and better understand the study and the factors evaluated for influencing stool quality.

Lines 362-391: belongs in Discussion

These lines were rewritten to speak of the results observed with discussion of the results moved to that section.

Lines 403-444: belongs in Discussion

These lines were rewritten to speak of the results observed with discussion of the results moved to that section.

Lines 445-458: belongs in Discussion

These lines were rewritten to speak of the results observed with discussion of the results moved to that section.

The Discussion and Conclusions should be rewritten to focus on the Objective. Furthermore, results of the study should be summarized in the Conclusions.

Discussion and conclusions were completely rewritten as requested.

Round 2

Reviewer 2 Report

Comments and Suggestions for Authors

The comments pertaining to the second review are in red and have been added to the authors responses provided to the initial review

Thank you for submitting your work for review. The manuscript attests to a large amount of work and data analysis. Unfortunately, the length of the manuscript does detract from its readability as it must be well over 10,000 words. I therefore recommend that the authors refine the manuscript to be more succinctly written with a tighter focus on the primary goal of the study with all additional information being supplementary. Alternatively, the manuscript could be split into more than one publication. 

We are very appreciative of the opportunity to present these data for consideration, and are grateful for your insightful and detailed comments.

I have provided an annotated PDF with my specific comments and queries which need to be addressed.

Thank you for this level of detail, it was exceptionally helpful. We have tried to address each of the points and feel that your review has strengthened the report. 

Much of my feedback was not addressed. It would have been useful for me undertaking the review of the revised manuscript to have had an explanation as to why some of the feedback was not addressed. If the reason is sensible then I would not provide the feedback again, however, without understanding why it has not been addressed I have restated many of my comments from the initial review round.

My general comments for each section are below.

Title: A simpler title would most likely engage more readers. For example: Nutrient digestive bypass: determinants and associations with stool quality in cats and dogs.

This was rewritten and the title simplified.  Thank you for addressing this

Abstract: The abstract is more than twice the length indicated in the authors' instructions which state "The abstract should be a total of about 200 words maximum." The results in the abstract lack precision.

This was rewritten and focused as requested. Thank you for addressing this

Introduction: The introduction covers sufficient information to provide context for the study. In places, supporting literature/references need to be cited. 

The references have been reworked.  However, we do not think that there is a need for references when the information is so general as to be included in multiple textbooks. There is no change in the reference citations in the introduction except for adjustment of one reference range from [6-11] to [6-10]

Methods: Clarification is required in several places as indicated in the annotated PDF. Some of the analyses need to be better justified. Much more detail is needed in the statistical analyses section (this can be provided as supplementary information). Each of the GLMs used should be detailed including the equation for the model.

We have read through the annotated PDF and made many of the suggested changes.  Very minimal changes have been made. For this round of revision please also returned the PDF with a response to each piece of feedback.

We do not think that repeating the actual detailed equations is valuable as the factors (i.e., the intercept and coefficients) are included in the tables. Fair enough however, the latter should be stated in the methods so that the reader knows the information will be presented in due course.

 Justification for the factors included is also required. For example, it is not clear why product type would be included in a model investigating nutrient bypass on stool quality especially if a preceding analysis has already examined the association of food type with nutrient bypass. Some of the models appear to be over-specified. Sufficient detail needs to be provided so that the same analysis can be repeated on a similar set of data.

We have included this justification in the methods.  The models were used because it was a retrospective study where these are the complete set of the factors available.   The models are directly represented in the tables so it is quite possible for other researchers to evaluate both the model and the coefficients from other data sets.

 Fair enough

Why were sex and whether or not the animal was desexed not considered/corrected for in the analyses? This requires an explanation. Given the effects of female sex hormones on intestinal motility and fluid balance, I would have thought this was important to factor in. 

We did assess the effect of sex as well as spay/neuter status as part of our assessment of factors which influence digestibility and stool firmness.  However, this assessment is fraught, since spay/neuter status is inherently confounded with age. Thus, there are no adult or older animals that are intact, and there are no puppies or kittens which were spayed/neutered. So the effect of spay/neuter status is inherently confounded with age. This is the reason we performed the analysis of the effect of age as a quadratic in our models so we could determine whether there were inflections in the effect of age at advanced age vs very young ages. To address the reviewer's point about hormonal influence of estrogens on fluid balance and GI function, we can look for differences by sex in unspayed/un-neutered intact young animals. Our data show that there were no differences between young intact male and female animals (dogs or cats) in TTTD for fat or protein, nor were there any differences by sex in stool firmness or moisture in these young intact dogs and cats. We also assessed any sex differences in adult, spayed/neutered animals even though these differences would not be due to the hormonal influences described by the reviewer. There were no differences by sex in spayed/neutered adult dogs for TTTD for fat or protein, nor were there any differences by sex in stool firmness or moisture. Interestingly, in spayed/neutered cats there was a statistically significant, albeit small effect size, effect of sex on stool firmness and moisture such that female cats had approximately 9% higher subjective stool firmness and ~5% lower (relative moisture). Since this effect of sex seen in adult cats was not present in dogs, it could not be due to the influence of sex hormones and was of a small effect size, thus  we elected to not include the effect of sex in our statistical models for cats. One reason for this was to enable cross-species comparisons in the data by utilizing the same model for both species. If sex had been included as a factor for cats only, and only in spayed/neutered adults, then the models would have been incomparable across species. We have revised the draft manuscript to report these findings in the hopes that they may be of use to spur future research into possible effects of sex in adult, spayed/neutered cats. 

Thank you for including this information. Please reword the conclusion regarding the hormones influence of sex hormone more conservatively. It is an assumption that just because the effect was not seen in both cats and dogs that it does not exist. There are species differences in sex hormone production that could explain the species variation – for example cat are induced ovulators and have several estrus cycles a year (where estrogen is at very elevated levels) whereas dogs typically only have 1 or 2 estrous cycles. Given female cats are therefore more likely to be in estrus compared to dogs it is possible this caused the species difference seen here for sex.

Was any analysis of the outliers performed? Did the outlier have anything in common? For example, if they were a dog were they a non-Beagel? 

The outliers were examined for patterns that might provide insight into commonalities, but there were no apparent patterns. There were no species, breed, sex, age, or weight patterns that distinguished the outliers from the final data set. It would appear that the outliers were a result of analytical variation and error in analysis of the fecal samples, since if the analytical error had occurred in the food then it would have impacted the entire panel of dogs on that particular digest test. So, we conclude there were no patterns that unified the outliers.

The analysis of outliers should be included as supplementary information and not just addressed in this forum. This information could be added under Supplementary Table S1.

Given the association of intestinal health with the microbiota there was an opportunity here to also investigate the microbial presence in each of the stool samples. Was this done?

Due to the retrospective nature of this historical data, samples were not available for microbiome analysis. We do appreciate the reviewer's comment about the importance of the microbiome on total tract digestibility and stool firmness, and we are performing followup interventional studies which will assess microbiome determinants of digestibility and stool quality.

 I wish you well in this research and look forward to reading about it. In addition to biomes (and the digestive functions), it would also be interesting to explore the expression levels of digestive proteins to gain more comprehensive insight into factors affecting total tract digestibility and stool firmness. Enzyme profiles would be expected to be different both between and within species.

For any given food type with a species were there enough animals tested at different ages to carry out a repeated measures ANOVA to verify any age effects detected with the GLM?

We interpret food type here to mean wet or dry food.  We interpret the request to ask whether there were enough animals at different ages to examine the effect of varying age on nutrient digestibility, nutrient intake, stool moisture and stool firmness within wet food only and similarly within dry food only.  While we have explored the effect of age itself on the range of endpoints included in the paper (for example, Table 2 summarizes nutrient digestibility of foods by species and food type.  Tables 5 and 6 summarize the effect of age on digestibility and nutrient intake, respectively, while Table 8 summarizes the effect of age on stool moisture and firmness), we have not explored the effect of age on nutrient digestibility, nutrient intake, stool moisture and stool firmness within a particular food type such as wet or dry.  It is true that we observed differences in protein, fat, NFE and fiber digestibility by food type though the patterns were not always consistent between cats and dogs (Section 3.3). We appreciate the reviewer’s suggestion while recognizing that we may need to balance the size and scope of the present manuscript to maintain the paper’s focus.  That said, a deeper exploration of the effect of age within a given food type may be potentially interesting to explore in future work. 

I realise the effect of age was explored, however, a repeated measures would have increased the statistical power so I was curious if it could be performed.

Results: The results need to be more focused on the main topic of the article. There are a lot of supplementary analyses presented in the main article which detract from the key results that align with the focus of the article. The focus should be on the models that determine the factors that influence nutrient bypass and the models that determine the factors associated with stool quality.

Thank you for this feedback.  We agree that the results section could be streamlined and have removed material not related to the core points of the paper from the results section to improve readability.

Nothing appears to have been removed from the results section in the revised paper I have recieved. Perhaps the incorrect revision has been uploaded? Please send a revision indicating what has been removed. Because the wrong version appears to have been uploaded for revision I have selected reconsider after major revisions so that I can be sent the correct revision.

Tables need to be formatted according to the authors' guidelines and not be a snip of the spreadsheet. The authors' guidelines state " Authors should use the Table option of Microsoft Word to create tables." Given the number of tables and how large some of them are, I would recommend moving the larger more complex tables to supplementary information as the key results are written in the text.

We have reworked and reformatted the tables.  However,  we think it important to allow the viewer to have this information in the main paper and we have left them in the manuscript.

Thank you for addressing this

Discussion: The discussion could be more concisely written if the re-iteration of results and speculation was removed. A section on the limitations of the study needs to be added.

We agree that the discussion section could be written more concisely and have removed material not related to the core points of the paper from this section to improve readability.   We have added a section on limitations as requested.

Again nothing appears to have been removed from the. The manuscript is still more than 10000 words. If material has been removed that should be indicated in the as text with a strikethrough in the revised version.

A brief section on limitations has been added, however, this section could have been written in more depth.

References: The list well exceeds the maximum of 30 as specified in the authors' instructions. The formatting of the references lacks consistency.

We have corrected the formatting.  We have surpassed the suggested maximum but feel the citations are required to adequately place these results with the previously published literature.

While I do not have an issue with the number of references cited it does not adhere to the author’s instructions so will be permitted at the editor's discretion

Supplementary Tables - The tables need informative titles and explanatory captions. The word "Removel" needs to be spelled correctly. Abbreviations need to be defined and units added where appropriate.

 We have corrected the tables as requested.

Abbreviation have not been defined as requested. Please correct the spelling of “effect” in Table S3

Author Response

Please find the attached file where we specifically speak to each of your comments.

Reviewer 4 Report

Comments and Suggestions for Authors

This manuscript describing the meta-analysis investigation of bypass nutrients impacts on stool firmness is well written. The authors have addressed previous concerns.

Author Response

Thank you for your careful review.  We appreciate that you now find "The authors have addressed previous concerns."

Round 3

Reviewer 2 Report

Comments and Suggestions for Authors

Only minor comments. Please see attached PDF. All the best fr your future research.

Comments on the Quality of English Language

The manuscript requires a final thorough proof read to pick up any remaining minor English language errors.

Author Response

Thank you again for your careful consideration and comments.  I think we have adequately addressed these last issues:

Line 24  the reviewer requested a concluding statement - the abstract was rewritten to include this statement.

Line 37 (in new document line 39) the reviewer requested a deletion of the incorrect period – this was done.

Line 99 and 102 - Usually supplementary tables and figures are labeled S1, S2 etc...

These were changed as requested.

Line 287 The justification for using this age needs to be provided. There is not physiological reason I can think of...

This justification was enhanced as requested.  The choice was the authors and chosen to supply the need for sufficient numbers and a younger group.  It was not based on a specific known physiological change but provides the reader with the observation of the change from this age which was the first quarter of the average lifespan.